# Towards Faithful Reasoning in Comics for Small MLLMs

## Abstract

Comic-based visual question answering (CVQA) poses distinct challenges to multimodal large language models (MLLMs) due to its reliance on symbolic abstraction, narrative logic, and humor, which differ from conventional VQA tasks. Although Chain-of-Thought (CoT) prompting is widely used to enhance MLLM reasoning, surprisingly, its direct application to CVQA often degrades performance, especially in small-scale models. Our theoretical and empirical analyses reveal that standard CoT in CVQA suffers from state entanglement, spurious transitions, and exploration inefficiency, with small models particularly vulnerable in resource-constrained settings. To address these issues, we propose a novel comic reasoning framework, designed to produce more faithful and transferable reasoning chains in small MLLMs. Specifically, our framework combines modular CoT generation with GRPO-based reinforcement fine-tuning and a novel structured reward. Experiments on three comic VQA benchmarks show that our method outperforms state-of-the-art models by an average of $10.4\%$ (up to $15.2\%$). When used as a plug-in component, it further yields an average improvement of $12.1\%$ across different MLLMs.

## 1 Introduction

Comics require layered reasoning over symbolic cues, cultural references, and narrative flow, making comic-based visual question answering (CVQA) substantially more challenging than conventional VQA. While multimodal large language models (MLLMs) achieve strong results on standard benchmarks, recent studies show that their performance on CVQA remains limited (Hu et al., 2024; Yang et al., 2024; Zhang et al., 2025; Liu et al., 2024), particularly for small-scale models that are widely used in practice. This gap underscores the need for methods that can strengthen reasoning under such challenging settings.

Chain-of-Thought (CoT) prompting (Wei et al., 2022) has emerged as a popular technique to enhance reasoning by encouraging intermediate steps (Wang et al., 2025; Li et al., 2025). However, in symbolically rich and context-dependent domains like CVQA, its effectiveness is far from guaranteed. Our experiments on CII-Bench (Zhang et al., 2025) reveal a counterintuitive result: *naive CoT prompting often degrades performance*, with small MLLMs suffering the most severe drop (Figure 1(A)). Since such lightweight models are central to resource-constrained deployments, this work focuses on understanding and improving the reasoning behavior of small MLLMs.

To shed light on why naive CoT degrades small MLLMs, we conduct a case study on Qwen2.5-VL-3B (Team, 2025b), a representative model. Our findings show that three undesirable patterns often emerge: 1). satirical target confusion—misidentifying the object of satire, 2). symbolic misalignment—misinterpreting culturally loaded symbols, and 3). salient cue omission—overlooking critical visual signals (Figure 2). These findings indicate that CoT can produce linguistically well-formed yet semantically unfaithful reasoning, echoing the phenomenon of verbal overshadowing (Liu et al., 2025), where explicit verbalization impairs perceptual judgment. As discussed in (Sprague et al., 2025), CoT mainly benefits formal symbolic reasoning tasks but can harm context-dependent, non-symbolic reasoning such as CVQA.

These observations raise a central question:

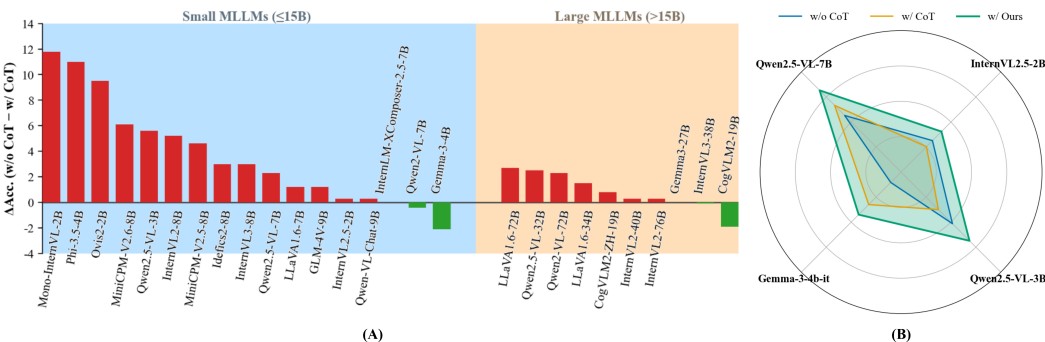

Figure 1: (A) Accuracy change with CoT prompting on CII-Bench, where naive CoT consistently degrades performance, with small MLLMs suffering larger drops and greater instability. The complete numerical results are provided in Appendix D. (B) Our plug-in consistently improves accuracy across small MLLMs on DEEPEVAL, compared with both w/ CoT and w/o CoT baselines.

> why does standard CoT, which succeeds in many reasoning tasks, fail so dramatically for small MLLMs in comic-based VQA?

In Sect. 2.1.1, we model reasoning as a sequential decision process and show that standard CoT suffers from three structural flaws: **state entanglement**, **spurious transitions**, and **exploration inefficiency**. **Small MLLMs are especially vulnerable**, since their limited capacity magnifies entanglement and makes them less robust to spurious trajectories—explaining the pronounced degradation observed in Figure 1. These insights motivate our proposed comic reasoning framework, which explicitly mitigates these flaws by enforcing modular reasoning and aligning optimization objectives with task-specific rewards, thereby producing more faithful and transferable CoTs.

**Our contributions are threefold:** (1) We provide the first systematic analysis of why standard CoT fails in comic VQA, bridging empirical failure patterns with a formal sequential decision perspective; (2) We introduce a new framework consisting of a modular and task-aligned CoT framework that enhances both faithfulness and transferability of reasoning in small MLLMs; (3) We achieve state-of-the-art results on three challenging comic-VQA benchmarks, with our 3B model outperforming baselines up to 7B, while our plug-in experiments further demonstrate model-agnostic gains across small MLLMs (Figure 1(B)).

## 2 METHOD

To address the limitations of standard CoT in comic VQA, we propose a novel two-stage framework: (i) **Mo**dular **C**hain-**of**-**T**hought generation (**MoCoT**) and (ii) reinforcement fine-tuning with **V**erifiable **E**nhanced **R**ew**A**rd (**VERA**), implemented via Group Relative Policy Optimization (GRPO (Shao et al., 2024)). MoCoT produces high-quality rationales, which are then used to supervise MLLM fine-tuning under VERA-guided reinforcement.

### 2.1 MODULAR CHAIN-OF-THOUGHT REASONING FOR VISUAL COMICS

We denote a CVQA instance as $\mathcal{I} = (I, Q)$, where $I$ is a comic image and $Q$ is the associated question. A reasoning trajectory is represented as $\tau = (z_1, \ldots, z_T)$, where each $z_t \in \mathcal{Z}$ is a latent reasoning state (e.g., grounding a visual cue, interpreting a symbolic reference, or inferring narrative flow). Reasoning is modeled as a policy $\pi$ over the state space $\mathcal{Z}$: $z_t \sim \pi(z_t \mid \mathcal{I}, z_{<t}), \ z_t \in \mathcal{Z}$.

#### 2.1.1 WHY STANDARD CoT FAILS IN COMIC VQA

**Motivation.** Unlike conventional VQA, CVQA requires reasoning over symbolic abstraction, narrative coherence, and humor. This makes reasoning chains highly context-dependent and error-prone. Naive CoT reasoning is subject to three structural flaws, which are summarized below.

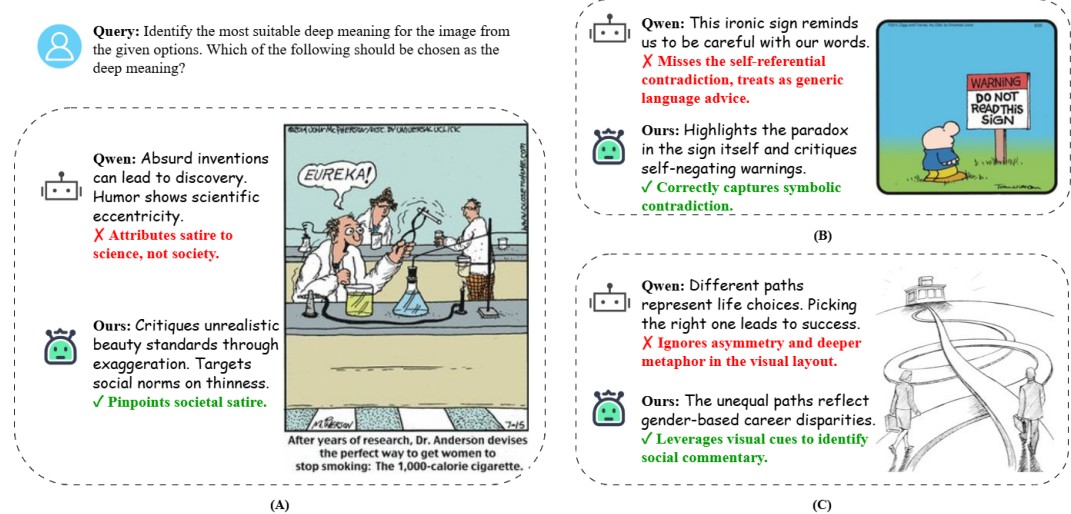

Figure 2: Representative failure cases of Qwen2.5-VL-3B (shown as Qwen in the figure) under naive CoT prompting. Typical errors include (A) satirical target confusion, (B) symbolic misalignment, and (C) salient cue omission, which directly lead to performance degradation. Our approach mitigates all the three factors.

**Proposition 2.1** (Limitations of Naive CoT). *Given a trajectory $\tau = (z_1, \ldots, z_T)$, naive CoT in CVQA exhibits: (i) **State entanglement**, each $z_t$ jointly encodes perceptual and abstract variables, preventing separation of error sources; (ii) **Spurious transitions**, since $\pi$ assigns non-zero probability to irrelevant symbolic states in $\mathcal{Z}$; and (iii) **Exploration inefficiency**, as the trajectory space $|\mathcal{T}| = |\mathcal{Z}|^T$ grows exponentially in $T$, making valid reasoning paths exponentially rare.*

*Remark.* These flaws are more evident in small MLLMs: their limited capacity amplifies entanglement, reduces robustness to spurious paths, and makes inefficient exploration particularly harmful—consistent with the empirical degradation observed in Figure 1(A). A complete proof is provided in Appendix B.1.

**Human Intuition.** When humans read comics, before consolidating them into a coherent judgment, we naturally factorize our reasoning into visual grounding, symbolic decoding, and narrative inference. MoCoT mirrors this strategy by enforcing modular reasoning steps that are auditable and verifiable.

### 2.1.2 MoCoT Pipeline Overview

MoCoT instantiates this idea as a three-stage *plan–execute–verify* pipeline (Figure 3):

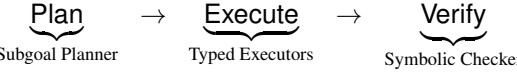

**Step 1: Subgoal Planning.** A planner $\mathcal{P}$ decomposes $(I, Q)$ into $K$ typed sub-questions: $\mathcal{Q}_{\text{sub}} = \{(q_k, t_k)\}_{k=1}^K$, $t_k \in \{\text{VISUAL}, \text{SYMBOLIC}, \text{NARRATIVE}\}$. Typing restricts the reasoning state space $\mathcal{Z}_{t_k} \subseteq \mathcal{Z}$, yielding focused sub-problems.

**Step 2: Localized Execution.** Each executor $\mathcal{E}_k$ independently solves its sub-question: $(r_k, a_k) = \mathcal{E}_k(I, q_k; t_k)$, producing localized rationales $r_k$ and provisional answers $a_k$. This results in a pool of sub-results $\mathcal{C}_{\text{sub}} = \{(r_k, a_k, t_k)\}_{k=1}^K$.

**Step 3: Meta-Reasoning and Verification.** A meta-reasoner consolidates the evidence into a diagnostic rationale (DTR) and a final inference rationale (FIR): $\text{DTR} = \text{Diagnose}(\mathcal{C}_{\text{sub}}, I, Q)$, $(\text{FIR}, A_o) = \text{Infer}(I, Q; \text{DTR})$. A symbolic checker $\mathcal{V}$ then validates entailment: $A'_o = \mathcal{V}(\text{FIR})$, accept iff $A'_o = A_o$.

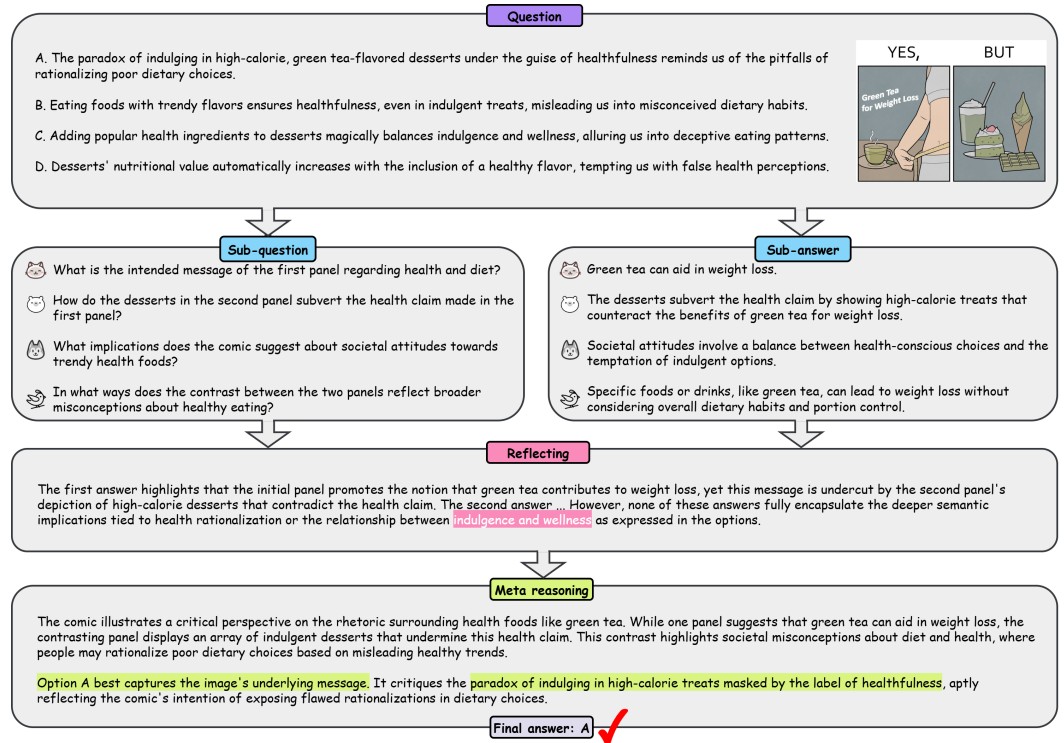

Figure 3: Our proposed MoCoT pipeline decomposes comic-based VQA tasks into structured sub-questions and sub-answers, followed by reflective reasoning and meta-level verification to guide final answer selection.

### 2.1.3 WHY MOCOT WORKS IN CVQA

MoCoT explicitly decomposes reasoning into $K$ sub-trajectories $\{\tau^{(k)}\}_{k=1}^{K}$, naturally aligning with the compositional structure of comic understanding.

**Definition 2.2** (Weak Subgoal Coupling). Consider a modular decomposition into $K$ sub-trajectories $\{\tau^{(k)}\}_{k=1}^{K}$, each governed by sub-policy $\pi_k$ over subspace $\mathcal{Z}_k \subseteq \mathcal{Z}$. We say subgoals are *weakly coupled* if $\max_{i \neq j} D_{\mathrm{KL}}\big(p(\tau^{(i)} \mid \tau^{(j)}, \mathcal{I}) \parallel p(\tau^{(i)} \mid \mathcal{I})\big) \leq \epsilon$, for some small $\epsilon > 0$.

**Proposition 2.3** (Value Decomposition of MoCoT). *Under modular reasoning and weak coupling, the global value approximately factorizes as $V(\mathcal{I}) \approx \sum_{k=1}^{K} V^{(k)}(s_0^{(k)})$, where $V^{(k)}$ is the expected reward of module $k$ from its initial state $s_0^{(k)}$.*

*Remark.* This modular factorization mitigates the three drawbacks of naive CoT: **(i) Reduced entanglement**, since perception and abstraction are handled by distinct modules; **(ii) Fewer spurious transitions**, as each $\pi_k$ explores only within its designated $\mathcal{Z}_k$; **(iii) Improved exploration efficiency**, reducing search from $O(|\mathcal{Z}|^T)$ to $O(\sum_{k=1}^{K} |\mathcal{Z}_k|^{T_k})$. Formal proofs are elaborated in Appendix B.2.

## 2.2 REINFORCEMENT FINE-TUNING WITH VERA

We adopt Group Relative Policy Optimization (GRPO) as the underlying reinforcement learning algorithm and introduce a verifiable alignment reward ,VERA, to align model reasoning with the unique demands of comic VQA.

### 2.2.1 GROUP RELATIVE POLICY OPTIMIZATION (GRPO)

GRPO (Shao et al., 2024) estimates advantages by comparing the relative rewards of multiple outputs for the same input, thus eliminating the need for an explicit value function. This is particularly

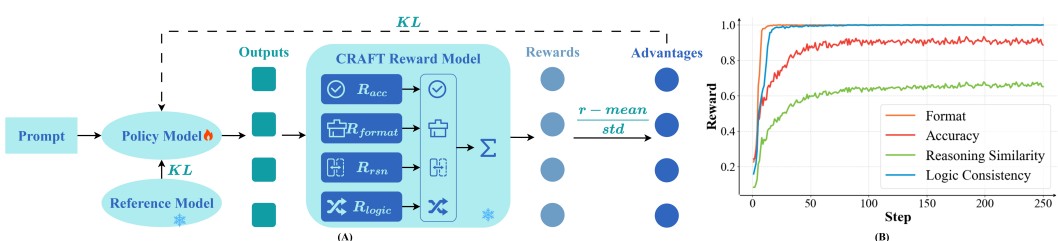

Figure 4: (A) Overview of GRPO with our proposed VERA reward function. Given a prompt, the policy model generates multiple outputs, which are scored by the VERA reward model. Rewards are normalized into group-relative advantages, and KL regularization ensures stability with respect to the reference model. (B) Our VERA reward trajectory during GRPO fine-tuning.

effective in tasks with sparse or delayed rewards, such as multi-step reasoning in CVQA. The clipped objective further regularizes towards a reference policy, ensuring stability while encouraging higher-reward outputs (Figure 4).

### 2.2.2 VERA: A STRUCTURED REWARD FOR VERIFIABLE REASONING

We propose **VERA**, which decomposes the reward into four interpretable dimensions: format correctness, answer accuracy, reasoning similarity, and logic consistency.

**Definition 2.4** (VERA Reward). For a generated output $o$, the structured reward is

$$R(o) = \lambda_1 R_{format}(o) + \lambda_2 R_{acc}(o) + \lambda_3 R_{rsn}(o) + \lambda_4 R_{logic}(o),$$

where $R_{format}$ checks structural compliance, $R_{acc}$ checks exact answer match, $R_{rsn}$ measures rationale similarity (activated only if $R_{acc} = 1$), and $R_{logic}$ enforces coherence between reasoning and conclusion.

*Remark.* VERA provides a balanced and interpretable training signal: $R_{format}$ enforces structural discipline, $R_{acc}$ ensures correctness, $R_{rsn}$ rewards semantically meaningful rationales, and $R_{logic}$ guarantees logical coherence. Together, these yield verifiable alignment, complementing MoCoT's modular reasoning.

Overall, our framework integrates two complementary innovations: (i) **MoCoT**, a modular reasoning pipeline that disentangles perceptual, symbolic, and narrative inference; and (ii) **VERA**, a structured reward that enforces verifiable reasoning. Their synergy enables small MLLMs to perform robustly on CVQA, addressing both reasoning efficiency and alignment. The complete pseudocode is provided in Appendix C.

## 3 EXPERIMENTS

### 3.1 EXPERIMENT SETTINGS

#### 3.1.1 DATASETS AND BENCHMARKS

To comprehensively evaluate our model's ability in comic-based reasoning, we employ three multiple-choice (MCQ) datasets: **DeepEval**, **YesBut v2** (referred to as YESBUT), and **CII-Bench**. DEEPEVAL (Yang et al., 2024) is designed to assess deep semantic understanding in comics, emphasizing high-level inference beyond surface-level recognition. YESBUT (Liang et al., 2025) extends the original YesBut benchmark (Hu et al., 2024) by introducing question samples constructed from semantically related panel pairs, where humor arises through contradictions or narrative twists. It covers a broad spectrum of reasoning complexity, ranging from literal comprehension to pragmatic inference. CII-BENCH (Zhang et al., 2025), by contrast, focuses on Chinese-language comics and culturally grounded visual-semantic understanding. It presents additional challenges due to linguistic differences and the prevalence of culturally specific humor rooted in Chinese society.

To construct data for MoCoT generation, we randomly sample 80% of the DEEPEVAL and YESBUT training sets (792 and 1,009 samples, respectively) to construct 930 high-quality CoT trajectories. Of

| Model | | | DEEPEVAL | YESBUT | CII-BENCH | | | | | | | | | |
|---|---|---|---|---|---|---|---|---|---|---|---|---|---|---|
| Name | #Params | CoT | | | Overall | Life | Art | Soc. | Pol. | Env. | CTC | Pos. | Neg. | Neu. |
| *7B Scale MLLMs* | | | | | | | | | | | | | | |
| LLaVA-1.6 | 7B | ✗ | 17.1 | 56.9 | 30.2 | 23.4 | 37.5 | 28.1 | 29.2 | 50.0 | 29.6 | 26.1 | 30.2 | 33.8 |
| | | ✓ | 29.7 | 54.9 | 29.0 | 21.7 | 34.6 | 30.3 | 29.2 | 44.4 | 28.2 | 25.6 | 30.2 | 30.8 |
| XComposer-2.5 | 7B | ✗ | 34.2 | 50.2 | 32.6 | 26.8 | 36.8 | 35.7 | 25.0 | 42.6 | 31.1 | 31.6 | 35.5 | 30.5 |
| | | ✓ | 36.2 | 45.5 | 32.6 | 30.3 | 32.4 | 34.6 | 33.3 | 40.7 | 30.4 | 31.6 | 35.1 | 30.8 |
| Qwen2.5-VL | 7B | ✗ | 58.3 | 68.8 | 48.1 | 41.1 | 52.2 | 51.4 | 58.3 | 53.7 | 47.4 | 47.9 | 47.2 | 49.3 |
| | | ✓ | 63.3 | 70.4 | 45.8 | 39.0 | 45.6 | 50.8 | 45.8 | 57.4 | 45.9 | 44.4 | 46.0 | 46.6 |
| InternVL3 | 8B | ✗ | 70.9 | 65.6 | 50.7 | 45.9 | 48.5 | 57.8 | 45.8 | 51.9 | 51.9 | 46.6 | 52.5 | 52.6 |
| | | ✓ | 67.8 | 66.4 | 47.7 | 42.9 | 46.3 | 55.1 | 37.5 | 57.4 | 45.2 | 46.2 | 47.9 | 48.9 |
| *≤4B Scale MLLMs* | | | | | | | | | | | | | | |
| Mono | 2B | ✗ | 14.1 | 48.2 | 22.5 | 17.8 | 22.8 | 21.1 | 29.2 | 27.8 | 28.9 | 23.1 | 21.1 | 23.3 |
| | | ✓ | 20.1 | 32.8 | 10.7 | 8.6 | 13.2 | 7.0 | 12.5 | 13.0 | 15.6 | 11.1 | 8.7 | 12.4 |
| Ovis2 | 2B | ✗ | 31.7 | 53.8 | 36.3 | 32.0 | 33.1 | 43.8 | 37.5 | 48.2 | 31.9 | 35.5 | 34.7 | 38.7 |
| | | ✓ | 32.2 | 50.6 | 26.8 | 22.1 | 28.7 | 35.1 | 37.5 | 35.2 | 16.3 | 23.1 | 28.3 | 28.6 |
| InternVL2.5 | 2B | ✗ | 45.7 | 45.5 | 33.6 | 27.3 | 36.8 | 37.3 | 41.7 | 40.7 | 31.9 | 30.8 | 34.7 | 35.0 |
| | | ✓ | 42.7 | 48.2 | 33.3 | 33.3 | 37.5 | 35.7 | 29.2 | 29.6 | 28.2 | 32.5 | 30.6 | 36.8 |
| Qwen2.5-VL | 3B | ✗ | 55.8 | 55.7 | 41.8 | 32.5 | 39.0 | 44.3 | 54.2 | 53.7 | 50.4 | 39.7 | 41.5 | 44.0 |
| | | ✓ | 48.7 | 57.7 | 36.2 | 31.2 | 33.8 | 34.6 | 37.5 | 50.0 | 43.7 | 37.2 | 31.7 | 39.9 |
| Phi-3.5 | 4B | ✗ | 35.7 | 56.9 | 33.1 | 26.8 | 39.0 | 32.4 | 45.8 | 44.4 | 31.9 | 26.5 | 37.4 | 34.6 |
| | | ✓ | 30.7 | 51.0 | 22.1 | 14.7 | 31.6 | 21.1 | 29.2 | 27.8 | 23.0 | 22.2 | 20.8 | 23.3 |
| Gemma-3 | 4B | ✗ | 35.2 | 51.0 | 30.5 | 26.8 | 34.6 | 31.4 | 45.8 | 35.2 | 26.7 | 23.5 | 34.3 | 32.7 |
| | | ✓ | 46.2 | 47.0 | 32.6 | 29.0 | 37.5 | 31.9 | 25.0 | 40.7 | 32.6 | 26.9 | 32.1 | 38.0 |
| **Ours** | 3B | – | **64.3**(+15.2%) | **62.9**(+9.0%) | **44.7**(+6.9%) | 35.9 | 44.1 | 49.2 | 50.0 | 55.6 | 48.9 | 41.0 | 44.9 | 47.7 |

Table 1: Overall accuracy (%) of different MLLMs (*with* ✓ and *without* ✗ CoT) and our method across three benchmarks: DEEPEVAL, YESBUT, and CII-BENCH[1] (evaluated by domains and emotions). The best and second-best results among ≤4B models are highlighted in **bold** and underlined, respectively.

these, 745 are used for GRPO fine-tuning, and 185 for validation. The remaining 20% of questions from DEEPEVAL (199 samples) and YESBUT (253 samples) are reserved for evaluation. The entire CII-BENCH dataset is used exclusively for validation.

### 3.1.2 IMPLEMENTATION DETAILS

All experiments are conducted on 4 NVIDIA A800 GPUs with 40GB of memory for each. For the MoCoT stage, all components are implemented using gpt-4o-mini (Hurst et al., 2024), except for the generation of diverse sub-answers, which is handled by Qwen2.5-VL-7B-Instruct (Team, 2025b).

For the GRPO stage, we adopt the EasyR1 (Zheng et al., 2025) framework with Qwen2.5-VL-3B-Instruct (Team, 2025b) as the base model. Fine-tuning is performed for 250 steps with a global batch size of 64. The rollout batch size is set to 256, with a tensor parallel size of 1. The VERA reward incorporates four components with empirically chosen weights: $\lambda_1 = 0.05$, $\lambda_2 = 0.6$, $\lambda_3 = 0.2$, and $\lambda_4 = 0.15$. The complete prompt design is provided in Appendix E.

[1]

### 3.2 MAIN RESULTS

**Benchmark Evaluation.** We first evaluate CVQA across three benchmarks—DEEPEVAL, YES-BUT, and CII-BENCH—under two regimes: w/o CoT (direct answering) and w/ CoT (reasoning-first). Table 1 reports all results.

Given the limited availability of established MLLM baselines for comic reasoning, we compare against a diverse set of strong models: (1) **LLaVA-v1.6-Mistral-7B** (Liu et al., 2023), (2) **InternLM-XComposer2.5-7B** (Zhang et al., 2024), (3) **Qwen2.5-VL-7B-Instruct** (Team, 2025b), (4) **InternVL3-8B** (Zhu et al., 2025), (5) **Mono-InternVL-2B** (Luo et al., 2024), (6) **Ovis2-2B** (Lu et al., 2024), (7) **InternVL2.5-2B** (Chen et al., 2024), (8) **Qwen2.5-VL-3B-Instruct** (Team, 2025b), (9) **Phi-3.5-Vision-Instruct** (Abdin et al., 2024), and (10) **Gemma-3-4b-it** (Team, 2025a).

---

[1]Soc.=Society, Pol.=Politics, Env.=Environment, CTC=Chinese Traditional Culture, Pos.=Positive, Neg.=Negative, Neu.=Neutral.

Across DEEPEVAL and YESBUT, our method consistently outperforms all ≤4B models under both w/ CoT and w/o CoT. Our 3B system attains 64.3% on DEEPEVAL—exceeding the 7B Qwen2.5-VL (63.3%)—and remains close to the 8B InternVL3 on YESBUT (ours 62.9% vs. 66.4%).

On CII-BENCH, which provides a finer-grained assessment by topical domains and sentiment, our approach achieves the best ≤4B overall accuracy (44.7%), ranks the first in 4/6 domains (Life, Art, Society, Environment), and leads in all three sentiment classes (Positive, Negative, Neutral), while remaining competitive on Politics and Chinese Traditional Culture.

From these results, it can be observed that both DEEPEVAL and CII-BENCH are more challenging than YESBUT, leading to lower absolute accuracies. Nevertheless, our framework maintains strong competitiveness, especially on CII-BENCH, where no training data was used—highlighting the potential of our method to generalize to unseen, fine-grained benchmarks. These results indicate that modular reasoning with reward-guided optimization scales robustly to diverse evaluation axes—even when built on compact models.

**Backbone-Agnostic Module Evaluation.** To further test the generality of our module, we attach it to four representative backbones (InternVL2.5-2B, Qwen2.5-VL-3B, Gemma-3-4B, Qwen2.5-VL-7B) and evaluate three settings on the DEEPEVAL dataset: w/o CoT, w/ CoT, and w/ Ours. For each backbone, the relative gain (Δ%) is computed against the *stronger* baseline between w/o and w/ CoT.

| Model | w/o CoT | w/ CoT | w/ Ours |
|---|---|---|---|
| InternVL2.5-2B | 45.7 | 42.7 | **50.3** (+10.1%) |
| Qwen2.5-VL-3B | 55.8 | 48.7 | **64.3** (+15.2%) |
| Gemma-3-4B | 35.2 | 46.2 | **51.3** (+11.0%) |
| Qwen2.5-VL-7B | 58.3 | 63.3 | **70.9** (+12.0%) |

Table 2: Backbone-agnostic evaluation. Accuracy (%) under *w/o* and *w/* CoT, and after adding our module. Δ% is computed against the stronger baseline.

As shown in Table 2, our method delivers consistent gains across 2–7B backbones. Averaged over the four models, it yields a **+12.1%** relative improvement. Concretely, the 3B backbone rises from 55.8/48.7 to 64.3 ( +15.2%), the 7B backbone reaches 70.9 ( +12.0%), and the 2B/4B backbones obtain +10.1% and +11.0% gains, respectively. These results underscore that our module is particularly effective in the low-parameter regime while remaining complementary to CoT prompting.

## 3.3 FURTHER ANALYSIS

### 3.3.1 QUALITATIVE COMPARISON

We present qualitative comparisons between our method and the baseline (Qwen2.5-VL-3B-Instruct) across three representative cases, each corresponding to a common failure pattern in comic reasoning. Complete qualitative examples are provided in Appendix F.

**(1) Symbolic Misalignment.** This failure pattern refers to the model's inability to interpret abstract metaphors or symbolic cues correctly. In a cartoon where a sign reads "DO NOT READ THIS SIGN," the baseline interprets the humor as a generic cautionary message, failing to recognize the self-referential contradiction. In contrast, our method successfully identifies the paradox and interprets the cartoon as a critique of performative or contradictory warnings, demonstrating stronger symbolic reasoning. (Corresponding to Figure 2(B))

**(2) Salient Visual Cue Omission.** This pattern captures cases where the model ignores or misreads critical objects or narrative signals in the image. For example, a cartoon shows two individuals taking divergent paths toward a tower, with clear visual asymmetry suggesting unequal difficulty. The baseline overlooks this and offers a vague interpretation about life choices. Our method, however, grounds its reasoning in the visual layout and correctly infers a commentary on gender-based disparity, highlighting better use of salient visual cues. (Corresponding to Figure 2(C))

**(3) Satirical Target Confusion.** This refers to the model detecting the presence of satire but misidentifying its intended target. In a cartoon featuring a "1,000-calorie cigarette," the baseline

attributes the humor to scientific absurdity, missing the deeper social critique. Our method correctly identifies the satire as a commentary on body image norms and gendered expectations, showing improved alignment with the cartoon's intended message. (Corresponding to Figure 2(A))

These cases illustrate that our approach better handles abstract symbolism, visual-grounded reasoning, and satire localization—three key aspects of deep comic understanding.

### 3.3.2 ABLATION STUDY

We conduct ablation experiments on the DEEPEVAL dataset to evaluate the impact of each component in our framework, including: (a) directly prompting the MLLM to generate CoTs and answers; (b) using only supervised fine-tuning (SFT) with MoCoT-generated data; (c) applying GRPO-based reinforcement fine-tuning directly on the MLLM with accuracy and format rewards; (d) GRPO fine-

| Setting | MLLM | MoCoT | GRPO | VERA | Acc. (%) |
|---------|------|-------|------|------|----------|
| (a) | ✓ | | | | 48.8 |
| (b) | ✓ | ✓ | | | 55.8 |
| (c) | ✓ | | ✓ | | 53.3 |
| (d) | ✓ | | ✓ | ✓ | 57.8 |
| (e) | ✓ | ✓ | ✓ | | 60.3 |
| (f) | ✓ | ✓ | ✓ | ✓ | **64.5** |

Table 3: Ablation study. Each row (a)–(f) corresponds to one experimental setting.

tuning with the VERA reward but without CoT supervision (i.e., removing the reasoning-similarity term); (e) GRPO fine-tuning with MoCoT data but using accuracy-only rewards; (f) our full framework, which applies GRPO fine-tuning with MoCoT data and the complete VERA reward.

As shown in Table 3, removing modular CoT generation (a) leads to a sharp performance drop, confirming the crucial role of structured CoTs. Omitting RL fine-tuning (b) also substantially hurts performance, with SFT accuracy close to direct prompting, showing that supervised learning alone cannot capture the complexities of comic reasoning. GRPO without CoT supervision (c) brings only limited gains, while adding the VERA reward (d) yields further improvements, highlighting the value of multi-dimensional rewards. Using MoCoT with GRPO but only accuracy-based rewards (e) performs better than SFT or accuracy-free GRPO, yet still lags behind the full model. The complete framework (f) achieves the best results, validating the complementary contributions of CoT supervision, reinforcement optimization, and structured reward design.

### 3.3.3 EFFECTIVENESS OF THE REWARD FUNCTION

To validate the effectiveness of our reward design, we track the evolution of each reward component throughout GRPO on the CVQA task.

As shown in Figure 4(B), all components exhibit consistent upward trends, demonstrating their effectiveness in shaping model behavior. Accuracy increases rapidly in the early stages, driven by its dominant weight, enabling the model to efficiently learn to produce correct answers. Reasoning similarity, which encourages alignment with human-authored CoTs, steadily improves from 0.08 to over 0.67, indicating enhanced capacity for structured and faithful inference. Meanwhile, logic consistency and format correctness also improve in tandem, promoting coherence and fluency in the generated reasoning chains.

These results confirm that our reward function effectively optimizes both factual correctness and reasoning quality, which are essential for success in complex multimodal tasks like comic understanding.

## 4 RELATED WORK

**CoT Reasoning in LLMs.** Chain-of-Thought (CoT) prompting has become a core technique for improving multi-step reasoning in large language models (LLMs). Early work introduced few-shot prompting using hand-crafted examples (Wei et al., 2022), but relied heavily on prompt engineering. Zero-shot CoT (Kojima et al., 2022) mitigated this by using simple trigger phrases (e.g., "Let's think step by step") to elicit reasoning without examples. Recent efforts have enhanced reasoning

quality and faithfulness. Multiagent Debate (Du et al., 2023) improved factual accuracy via inter-agent critique, while Process Supervision (Lightman et al., 2023) provided step-level feedback to train reward models. Question Decomposition (Radhakrishnan et al., 2023) improved robustness by solving sub-problems and linking them to final conclusions. Metacognitive Prompting (Bai et al., 2025) further integrated planning and reflection for lateral-thinking tasks. Together, these studies reflect a shift toward structured and cognitively inspired reasoning in LLMs.

**CoT Reasoning in MLLMs.**  Inspired by LLM advances and the success of DeepSeek-style reasoning (Shao et al., 2024; Guo et al., 2025), recent studies have extended CoT prompting to MLLMs, which face challenges like visual grounding, hallucination, and limited data. URSA (Luo et al., 2025) tackled these with a large-scale dataset (MMathCoT-1M) and a dual-perspective verifier for logic and vision. Vision-R1 (Huang et al., 2025) added reinforcement learning with modality bridging and verbosity control. Qwen-VL-DP (Shi et al., 2025) introduced multi-path reasoning with diversity-aware reward signals. These works advance verifiable and multi-perspective CoT in multimodal settings.

While previous approaches tend to focus on decomposition, critique, or supervision in isolation, our work integrates these components into a unified modular CoT framework, facilitating interpretable and semantically consistent reasoning.

**Comic-based VQA in MLLMs.**  Recent studies have explored whether MLLMs can capture the humor, satire, and implicit semantics of comics and memes. Early work introduced the New Yorker Humor Benchmark (Hessel et al., 2023), evaluating caption matching, ranking, and explanation tasks. MemeCap (Hwang & Shwartz, 2023) extended this to meme captioning, highlighting the difficulty of visual metaphor interpretation. Moving beyond surface humor, DeepEval (Yang et al., 2024) and II-Bench (Liu et al., 2024) assessed deep semantic and implicature understanding, showing large gaps between MLLMs and humans. Other benchmarks targeted specific structures, such as YESBUT for multi-panel juxtaposition (Hu et al., 2024) and CII-Bench for Chinese cultural contexts (Zhang et al., 2025). Together, these benchmarks underscore the unique challenges of CVQA and call for methods that can strengthen the reasoning ability of MLLMs in such settings.

Most recently, the LAD framework (Zhang & Niu, 2025) introduced perception–search–reasoning modules, narrowing the performance gap with commercial systems. However, both II-Bench and CII-Bench largely attributed the weaker performance of smaller MLLMs with CoT prompting to model scale, overlooking that CoT itself may degrade reasoning in CVQA—a gap our work directly addresses. Moreover, while LAD improves performance by retrieving external information, our focus is on unleashing the latent reasoning capacity of MLLMs without external augmentation, particularly under resource-constrained settings where scaling up is not feasible.

## 5 CONCLUSION

This work highlights a central paradox in multimodal reasoning: while Chain-of-Thought prompting is celebrated for enhancing reasoning in many domains, it can backfire in comic-based VQA, especially for small MLLMs. Our analysis shows that the challenges of symbolic ambiguity, cultural grounding, and narrative complexity make comics a unique stress test where naive CoT often produces fluent but unfaithful reasoning.

In response, we introduced a new framework for comics that rethinks how reasoning should be structured for compact multimodal models. Instead of scaling parameters, our method emphasizes modular decomposition, interpretable intermediate steps, and reward-aligned optimization. This design allows small models to not only close the gap with, but in some cases surpass, larger counterparts on multiple challenging benchmarks.

More broadly, our findings suggest that effective reasoning in multimodal contexts requires structure, not just scale. By exposing the limits of standard CoT and demonstrating a path forward, this work points toward a new agenda: building reasoning frameworks that generalize robustly across symbolic, cultural, and perceptual dimensions. Future directions include adaptive reward shaping and applying our method beyond comics to other domains where reasoning fidelity is critical.

## ETHICS STATEMENT

Our work does not involve any human subjects, sensitive data, or applications with potential ethical risks. Moreover, this work raises no known ethical concerns.

## REPRODUCIBILITY STATEMENT

To ensure reproducibility, we have provided an anonymized package at the anonymous link[2], which contains both the implementation and the train/validation datasets. Details of model architectures, hyperparameters, and training procedures are described in Sect. 3.1.2, and all theoretical assumptions and complete proofs are presented in Appendix B.

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

APPENDIX

## A    USAGE OF LARGE LANGUAGE MODELS (LLMS)

We used Large Language Models (LLMs) only for polishing writings, and grammar checking. No LLMs were involved in designing experiments, analyzing data, or contributing to the scientific findings of this work.

## B    DETAILED PROOFS

### B.1    PROOF OF THEOREM 2.1.1

**Setup.**    Let a CVQA instance be $\mathcal{I} = (I, Q)$, where $I$ is the comic image (possibly multi-panel) and $Q$ is the associated question. A reasoning trajectory is $\tau = (z_1, \ldots, z_T)$ with states $z_u \in \mathcal{Z}$, and we denote the prefix by $z_{<u} = (z_1, \ldots, z_{u-1})$. The policy is $\pi_\theta(z_u \mid \mathcal{I}, z_{<u})$, parameterized by $\theta$, assigning probabilities over $\mathcal{Z}$. For analysis we decompose each state as

$$z_u = (z_u^{\mathrm{perc}}, z_u^{\mathrm{abs}}), \qquad \mathcal{Z} = \mathcal{Z}_{\mathrm{perc}} \times \mathcal{Z}_{\mathrm{abs}},$$

where $z_u^{\mathrm{perc}}$ captures perceptual variables and $z_u^{\mathrm{abs}}$ captures abstract/narrative variables. We write $f_\theta(z; \mathcal{I}, z_{<u}) \in \mathbb{R}$ for the logit score of state $z$, so that

$$\pi_\theta(z \mid \mathcal{I}, z_{<u}) = \frac{\exp f_\theta(z; \mathcal{I}, z_{<u})}{\sum_{z' \in \mathcal{Z}} \exp f_\theta(z'; \mathcal{I}, z_{<u})}.$$

Let $\mathcal{Z}_{\mathrm{sym}} \subset \mathcal{Z}$ denote symbolic states that are irrelevant to answering $Q$ under $\mathcal{I}$. The abstract component is assumed to couple with perceptual cues through a noisy mapping $z_u^{\mathrm{abs}} = g(z_u^{\mathrm{perc}}, \mathcal{I}, z_{<u}, \varepsilon)$, where $\varepsilon$ is an exogenous noise independent of $(\mathcal{I}, z_{<u})$ with $\mathrm{Var}(\varepsilon) > 0$.

Validity of a trajectory is encoded by the indicator $\mathbf{1}_{\mathrm{valid}}(\tau) \in \{0, 1\}$, equal to 1 iff $\tau$ is a correct reasoning path. We denote the valid set $\mathcal{T}_{\mathrm{valid}} \subseteq \mathcal{Z}^T$ and its fraction $\rho_T = |\mathcal{T}_{\mathrm{valid}}| / |\mathcal{Z}|^T$. For stepwise reasoning we also define $V_u^{\mathrm{glob}}(\mathcal{I}, z_{<u}) \subseteq \mathcal{Z}$ as the set of valid next states. We assume there exists a constant $\overline{p}_{\mathrm{glob}} < 1$ such that the probability mass assigned by $\pi_\theta$ to valid next states is at most $\overline{p}_{\mathrm{glob}}$, and their relative size satisfies $|V_u^{\mathrm{glob}}| \leq \kappa|\mathcal{Z}|$ for some $\kappa \in (0, 1)$.

**Lemma B.1** (State entanglement is generic). *Under the setup above, for almost every $(\mathcal{I}, z_{<u})$ one has*

$$p(z_u \mid \mathcal{I}, z_{<u}) \neq p(z_u^{\mathrm{perc}} \mid \mathcal{I}, z_{<u}) \, p(z_u^{\mathrm{abs}} \mid \mathcal{I}, z_{<u}).$$

*Hence $z_u^{\mathrm{perc}}$ and $z_u^{\mathrm{abs}}$ fail to be conditionally independent given $(\mathcal{I}, z_{<u})$, and entanglement is unavoidable in general.*

*Proof.* By construction, $z_u^{\mathrm{abs}} = g(z_u^{\mathrm{perc}}, \mathcal{I}, z_{<u}, \varepsilon)$ with non-degenerate $\varepsilon$. Therefore the conditional law of $z_u^{\mathrm{abs}}$ depends on $z_u^{\mathrm{perc}}$ (via $g$) unless $g$ is a.e. constant in its first argument, which contradicts the comic-narrative coupling. Thus $p(z_u^{\mathrm{abs}} \mid z_u^{\mathrm{perc}}, \mathcal{I}, z_{<u}) \neq p(z_u^{\mathrm{abs}} \mid \mathcal{I}, z_{<u})$ almost everywhere, implying

$$p(z_u \mid \mathcal{I}, z_{<u}) = p(z_u^{\mathrm{abs}} \mid z_u^{\mathrm{perc}}, \mathcal{I}, z_{<u})\, p(z_u^{\mathrm{perc}} \mid \mathcal{I}, z_{<u}) \neq p(z_u^{\mathrm{abs}} \mid \mathcal{I}, z_{<u})\, p(z_u^{\mathrm{perc}} \mid \mathcal{I}, z_{<u}).$$

$\square$

**Lemma B.2** (Inevitable spurious transitions). *Under the setup above, for any $(\mathcal{I}, z_{<u})$ and any $\theta$ not lying in a measure-zero set,*

$$\sum_{z \in \mathcal{Z}_{\mathrm{sym}}} \pi_\theta(z \mid \mathcal{I}, z_{<u}) > 0.$$

*Therefore trajectories drawn from $\pi_\theta$ admit spurious moves into $\mathcal{Z}_{\mathrm{sym}}$ with strictly positive probability.*

*Proof.* By softmax positivity, $\pi_\theta(z \mid \cdot) > 0$ iff $f_\theta(z; \cdot)$ is finite; in standard neural parameterizations, logits are finite almost everywhere in $\theta$. Since $|\mathcal{Z}_{\mathrm{sym}}| \geq 1$, it suffices to show existence of at least one $z \in \mathcal{Z}_{\mathrm{sym}}$ with $\pi_\theta(z \mid \cdot) > 0$. Because $f_\theta$ is continuous in $\theta$ and typically non-constant across $z$, the set of parameters enforcing *exact zeros* on a prescribed subset is a measure-zero manifold. Thus for almost all $\theta$, each $z \in \mathcal{Z}$ receives strictly positive mass. Summing over $\mathcal{Z}_{\mathrm{sym}}$ yields the claim. $\square$

**Lemma B.3** (Exploration complexity and exponential rarity). *For trajectory length $T$, the probability that a trajectory sampled from $\pi_\theta$ is valid satisfies*

$$\mathbb{P}_\pi(\tau \in \mathcal{T}_{\mathrm{valid}}) \leq \overline{p}_{\mathrm{glob}}^T \leq \big( \max\{\kappa, \overline{p}_{\mathrm{glob}}\} \big)^T,$$

*which decays exponentially in $T$. In particular, if $|\mathcal{T}_{\mathrm{valid}}| \leq (\kappa|\mathcal{Z}|)^T$ for some $\kappa < 1$, then under uniform sampling the success probability is $\rho_T = \Theta(\kappa^T)$.*

*Proof.* A valid trajectory must pick a state in $V_u^{\mathrm{glob}}$ at each step. By the law of total probability and the per-step bound,

$$\mathbb{P}_\pi(\tau \in \mathcal{T}_{\mathrm{valid}}) = \mathbb{E}\Big[ \prod_{u=1}^{T} \sum_{z \in V_u^{\mathrm{glob}}} \pi_\theta(z \mid \mathcal{I}, z_{<u}) \Big] \leq \prod_{u=1}^{T} \overline{p}_{\mathrm{glob}} = \overline{p}_{\mathrm{glob}}^T.$$

Since $|V_u^{\mathrm{glob}}|/|\mathcal{Z}| \leq \kappa$ and the uniform policy achieves $\kappa$ per-step mass, we also have $\overline{p}_{\mathrm{glob}} \leq \max\{\kappa, \overline{p}_{\mathrm{glob}}\}$, giving the second inequality. For the uniform sampler, $\mathbb{P}_{\mathrm{unif}}(\tau \in \mathcal{T}_{\mathrm{valid}}) = \rho_T = |\mathcal{T}_{\mathrm{valid}}|/|\mathcal{Z}|^T$; if $|\mathcal{T}_{\mathrm{valid}}| \leq (\kappa|\mathcal{Z}|)^T$ then $\rho_T \leq \kappa^T$. $\square$

**Corollary B.4** (Proof of Theorem 2.1.1). *By Lemma B.1, naive CoT induces unavoidable entanglement between perceptual and abstract factors. By Lemma B.2, softmax policies necessarily assign nonzero probability to irrelevant symbolic states, inducing spurious transitions. By Lemma B.3, the probability of sampling a valid trajectory without additional structure decays exponentially in $T$. Therefore standard CoT in CVQA suffers simultaneously from state entanglement, spurious transitions, and exploration inefficiency.* $\square$

**Remarks on tightness.** The bounds in Lemma B.3 are tight up to constants: if per-step valid sets occupy at most a fraction $\kappa < 1$ of the state space and the policy mass on them is bounded by $\overline{p}_{\mathrm{glob}}$, then the best-case success probability is at most $\overline{p}_{\mathrm{glob}}^T$; under uniform sampling it matches $\rho_T$. Moreover, Lemma B.2 can be strengthened to show that suppressing *all* spurious states requires measure-zero parameter choices (degenerate logits), which is unstable under training perturbations.

## B.2 PROOF OF THEOREM 2.1.3

**Setup (inherits from Appendix B.1).** We reuse the CVQA instance $\mathcal{I} = (I, Q)$, the state space $\mathcal{Z} = \mathcal{Z}_{\text{perc}} \times \mathcal{Z}_{\text{abs}}$, the (global) trajectory $\tau = (z_1, \ldots, z_T)$, and the set of symbolic-irrelevant states $\mathcal{Z}_{\text{sym}} \subset \mathcal{Z}$. MoCoT replaces the single policy $\pi_\theta$ with a modular *plan–execute–verify* pipeline:

$$\text{Plan} \to \text{Execute} \to \text{Verify}.$$

Planning yields $K$ typed sub-questions $\{(q_k, t_k)\}_{k=1}^K$ with types $t_k \in \{\text{VISUAL}, \text{SYMBOLIC}, \text{NARRATIVE}\}$. Each type induces a typed subspace $\mathcal{Z}_{t_k} \subseteq \mathcal{Z}$ and a sub-policy $\pi_{t_k}$ supported on $\mathcal{Z}_{t_k}$. Execution produces sub-trajectories $\tau^{(k)} = (z_1^{(k)}, \ldots, z_{T_k}^{(k)})$ with $z_s^{(k)} \in \mathcal{Z}_{t_k}$ and $\sum_{k=1}^K T_k = T$. A symbolic checker $\mathcal{V}$ accepts a composed rationale/answer iff it passes type-consistency and entailment checks.

**Notation guard (local to this subsection).** We reserve $t$ for *types* and $s$ for *module-internal* steps. Global valid sets from Appendix B.1 are $V_u^{\text{glob}}(\mathcal{I}, z_{<u})$ at global step $u$. Typed valid sets are $V_s^{(t)}(\mathcal{I}, z_{<s}^{(t)}) \subseteq \mathcal{Z}_t$. Branching factors: $B := |\mathcal{Z}|$, $B_t := |\mathcal{Z}_t|$. Let $\Delta$ be the type-interface ambiguity set and $\delta_{\text{type}} := |\Delta|/|\mathcal{Z}|$. For each type $t$,

$$\underline{p}_t := \inf_{s, \mathcal{I}, z_{<s}^{(t)}} \sum_{z \in V_s^{(t)}} \pi_t(z \mid \mathcal{I}, z_{<s}^{(t)}), \quad \overline{p}_t := \sup_{s, \mathcal{I}, z_{<s}^{(t)}} \sum_{z \in V_s^{(t)}} \pi_t(z \mid \mathcal{I}, z_{<s}^{(t)}), \quad \kappa_t := \sup_s \frac{|V_s^{(t)}|}{|\mathcal{Z}_t|}.$$

Verifier errors: $\alpha$ (false reject), $\beta$ (false accept).

**Assumptions (mild and modular).**

- **A1 (Typed support).** For each type $t$, $\text{supp}(\pi_t) \subseteq \mathcal{Z}_t$ and $\mathcal{Z}_t \cap \mathcal{Z}_{t'} = \emptyset$ for $t \neq t'$, except possibly on a negligible interface $\Delta$ with $\frac{|\Delta|}{|\mathcal{Z}|} \leq \delta_{\text{type}}$.

- **A2 (Weak subgoal coupling).** For the modular decomposition $\{\tau^{(k)}\}_{k=1}^K$,

$$\max_{i \neq j} D_{\text{KL}}\big(p(\tau^{(i)} \mid \tau^{(j)}, \mathcal{I}) \,\big\|\, p(\tau^{(i)} \mid \mathcal{I})\big) \leq \varepsilon.$$

- **A2′ (Typed latent mediator).** In the no-interface event $E^c$, there exists a typed latent mediator $S^{(t)}$ such that

$$X \leftarrow S^{(t)} \to Y \quad \text{given } (\mathcal{I}, z_{<s}^{(t)}, t, E^c),$$

and $I(S^{(t)}; \tau^{(-t)} \mid \mathcal{I}, z_{<s}^{(t)}, t, E^c) \leq \varepsilon$.

- **A3 (Verifier reliability).** With composed hypothesis $H$ (DTR/FIR + answer),

$$\mathbb{P}\big[\mathcal{V}(H) = 1 \mid H \text{ invalid}\big] \leq \beta, \quad \mathbb{P}\big[\mathcal{V}(H) = 0 \mid H \text{ valid}\big] \leq \alpha < \tfrac{1}{2}.$$

- **A4 (Module sparsity).** For each $t$, $\kappa_t = \sup_s |V_s^{(t)}|/|\mathcal{Z}_t| < 1$, and $\underline{p}_t \leq \sum_{z \in V_s^{(t)}} \pi_t(z \mid \cdot) \leq \overline{p}_t$ uniformly in $s$.

**Lemma B.5** (Typed disentanglement bounds). *Under A1, A2, and A2′, for any module of type $t$ and step $s$,*

$$I\left(z_{s,\text{perc}}^{(t)}; z_{s,\text{abs}}^{(t)} \,\Big|\, \mathcal{I}, z_{<s}^{(t)}, t\right) \leq \varepsilon + h(\delta_{\text{type}}),$$

*where one admissible choice is $h(\delta) = H_2(\delta) + \delta \log B_t$ with $H_2(\cdot)$ the binary entropy; $h$ is monotone and satisfies $h(0) = 0$.*

*Proof.* Let $C := (\mathcal{I}, z_{<s}^{(t)}, t)$, $X := z_{s,\text{perc}}^{(t)}$, $Y := z_{s,\text{abs}}^{(t)}$. Let $E$ be the "type-interface" event with $\delta := \mathbb{P}(E = 1 \mid C) \leq \delta_{\text{type}}$.

*Step 1 (Mixture by the interface).* By the chain rule of conditional MI and the definition of conditional interaction information,

$$I(X; Y \mid C) = (1 - \delta) I(X; Y \mid C, E^c) + \delta I(X; Y \mid C, E) + I(E; X; Y \mid C).$$

Since $|I(E; X; Y \mid C)| \leq H_2(\delta)$, we obtain

$$I(X; Y \mid C) \leq (1 - \delta) I(X; Y \mid C, E^c) + \delta I(X; Y \mid C, E) + H_2(\delta). \qquad (1)$$

*Step 2 (Interface term).* On $E$, type mixing can increase dependence but $X, Y$ take values in a finite typed subspace, hence $I(X; Y \mid C, E) \leq \log B_t$. With $\delta \leq \delta_{\text{type}}$, this contributes at most $\delta_{\text{type}} \log B_t$.

*Step 3 (Typed-subspace term via mediator).* In the event $E^c$, by **A2**$'$ there exists a typed mediator $S^{(t)}$ such that $X \leftarrow S^{(t)} \rightarrow Y$ given $(C, E^c)$ and $I(S^{(t)}; \tau^{(-t)} \mid C, E^c) \leq \varepsilon$. By information decomposition and data processing,

$$I(X; Y \mid C, E^c) \leq I(S^{(t)}; X \mid C, E^c) + I(S^{(t)}; Y \mid C, E^c) \leq \varepsilon.$$

*Step 4 (Combine).* Plugging these into equation 1 yields

$$I(X; Y \mid C) \leq \varepsilon + H_2(\delta_{\text{type}}) + \delta_{\text{type}} \log B_t = \varepsilon + h(\delta_{\text{type}}).$$

$\square$

**Lemma B.6** (Suppression of spurious symbolic states)**.** *Under **A1** and **A3**, the probability that the final MoCoT output involves any spurious move into $\mathcal{Z}_{\text{sym}}$ is at most*

$$\beta + K \delta_{\text{type}},$$

*where $K$ can be taken as $K \leq T$ (or $K \leq \sum_{k=1}^{K} T_k$).*

*Proof.* By **A1**, for $t \neq \text{SYMBOLIC}$ we have $\mathcal{Z}_{\text{sym}} \cap \mathcal{Z}_t = \emptyset$ (up to $\Delta$), so non-symbolic modules assign zero mass to $\mathcal{Z}_{\text{sym}}$ unless traversing $\Delta$. A union bound over at most $T$ steps gives probability at most $K \delta_{\text{type}}$. Symbolic content is handled within the SYMBOLIC module and then checked by $\mathcal{V}$; by **A3** spurious acceptance occurs with probability at most $\beta$. Summing gives $\beta + K \delta_{\text{type}}$. $\square$

**Lemma B.7** (Modular exploration efficiency)**.** *Under **A4**, each module $k$ of type $t_k$ satisfies*

$$\mathbb{P}\left( \tau^{(k)} \in \mathcal{T}_{\text{valid}}^{(k)} \right) \geq \underline{p}_{t_k}^{T_k}.$$

*Moreover,*

$$\mathbb{P}\left( \text{all modules valid} \right) \geq (1 - c\,\varepsilon) \prod_{k=1}^{K} \underline{p}_{t_k}^{T_k},$$

*for some constant $c > 0$ from weak coupling (**A2**). For uniform exploration in $\mathcal{Z}_{t_k}$, $\mathbb{P}_{\text{unif}}(\tau^{(k)} \in \mathcal{T}_{\text{valid}}^{(k)}) = \Theta(\kappa_{t_k}^{T_k})$.*

**Proposition B.8** (End-to-end success with verification)**.** *Under **A3** and Lemma B.7,*

$$\mathbb{P}\left( \text{MoCoT outputs a valid answer} \right) \geq (1 - \alpha)(1 - c\,\varepsilon) \prod_{k=1}^{K} \underline{p}_{t_k}^{T_k}.$$

**Theorem B.9** (Why MoCoT works in CVQA)**.** *Assume **A1–A4** and **A2**$'$. Let standard CoT satisfy the per-step bound of Lemma B.3 with parameter $\overline{p}_{\text{glob}}$ and valid fraction $\kappa$. Then MoCoT yields:*

1. ***Entanglement reduction:*** *By Lemma B.5, within-module dependence is bounded by $\varepsilon + h(\delta_{\text{type}})$, strictly smaller than generic entanglement.*

2. ***Spurious suppression:*** *By Lemma B.6, the spurious probability is at most $\beta + K\,\delta_{\text{type}}$, whereas standard CoT assigns positive mass to $\mathcal{Z}_{\text{sym}}$ almost surely.*

3. ***Exploration efficiency:*** *Standard CoT success $\leq (\max\{\kappa, \overline{p}_{\text{glob}}\})^T$; MoCoT achieves $\geq (1 - \alpha)(1 - c\varepsilon) \prod_k \underline{p}_{t_k}^{T_k}$. For uniform exploration, the search reduces from $\Theta(\kappa^T)$ to $\Theta(\prod_k \kappa_{t_k}^{T_k})$ with $B_{t_k} \ll B$.*

*Thus MoCoT mitigates state entanglement, spurious symbolic transitions, and exponential exploration hardness.* $\square$

**Remarks on tightness and design levers.**

- Lemma B.5 tightens as typing improves ($\delta_{\text{type}} \downarrow 0$) and subgoals decouple ($\varepsilon \downarrow 0$); in practice this means stronger Plan and cleaner DTR→FIR interfaces.

- Lemma B.6 shows that spurious probability is dominated by $\beta$; improving Verify (e.g., stricter consistency checks) directly reduces it.

- Exploration gains arise from smaller $B_t$ and larger $\underline{p}_t$, both compounding exponentially with $T_k$.

- Structured rewards (e.g., VERA) can further increase $\underline{p}_t$ and decrease $\beta$, improving both constants and exponential rates.

## C  ALGORITHM DESCRIPTION

We provide the pseudocode for the two components of our framework: (i) MoCoT for modular chain-of-thought generation (Algorithm 1), and (ii) VERA-guided GRPO fine-tuning for verifiable alignment (Algorithm 2).

---

**Algorithm 1** MoCoT

---

**Require:** Comic image $I$, question $Q$
**Ensure:** Final answer $A_o$ with validated rationale
1: Initialize modules: planner $\mathcal{P}$, executors $\{\mathcal{E}_k\}$, meta-reasoner, and checker $\mathcal{V}$
2: **repeat**
3:     $\mathcal{Q}_{\text{sub}} \leftarrow \mathcal{P}(I, Q)$                                       ▷ Decompose into $K$ typed sub-questions
4:     $\mathcal{Q}_{\text{sub}} = \{(q_k, t_k)\}_{k=1}^K,\ t_k \in \{\text{VISUAL}, \text{SYMBOLIC}, \text{NARRATIVE}\}$
5:     Restrict reasoning space: $\mathcal{Z}_{t_k} \subseteq \mathcal{Z}$ for each type $t_k$
6:     **for** $k = 1$ to $K$ **do**
7:         $(r_k, a_k) \leftarrow \mathcal{E}_k(I, q_k; t_k)$     ▷ Executor produces rationale $r_k$ and provisional answer $a_k$
8:     **end for**
9:     $\mathcal{C}_{\text{sub}} \leftarrow \{(r_k, a_k, t_k)\}_{k=1}^K$                      ▷ Pool of typed sub-results
10:     $\text{DTR} \leftarrow \text{Diagnose}(\mathcal{C}_{\text{sub}}, I, Q)$       ▷ Aggregate evidence into diagnostic rationale
11:     $(\text{FIR}, A_o) \leftarrow \text{Infer}(I, Q; \text{DTR})$     ▷ Generate final inference rationale and answer
12:     $A_o' \leftarrow \mathcal{V}(\text{FIR})$                ▷ Checker validates entailment of the final rationale
13: **until** $A_o' = A_o$
14: **return** $A_o$

---

## D  FULL RESULTS FOR FIGURE 1(A)

For completeness, we report the full numerical results corresponding to Figure 1(A), which illustrates the effect of naive CoT prompting on CII-BENCH. While the main paper shows the accuracy change in aggregate, Tables 4 and 5 provide the detailed results for Small and Large MLLMs, respectively. As can be seen, naive CoT prompting often leads to performance drops, especially for smaller models.

## E  PROMPT LIST

We provide the exact system prompts used in our experiments. Specifically, Table 6, 7, 8, and 9 correspond to the prompts for Step 1 (Subgoal Planning), Step 2 (Localized Execution), and Step 3 (Meta-Reasoning and Verification) in the MoCoT pipeline. In addition, Table 10 presents the system prompt used for VERA-guided GRPO fine-tuning, which enforces structured output formatting. Finally, Tables 11 and 12 provide the prompts employed in evaluating MLLMs without and with chain-of-thought reasoning, respectively.

---

**Algorithm 2** GRPO Fine-tuning with VERA Reward

---

**Require:** Initial policy $\pi_\omega^0$, dataset $\mathcal{D}$, reward functions $\{R_{format}, R_{acc}, R_{rsn}, R_{logic}\}$ with weights $\{\lambda_i\}$, hyperparameters: $N$ (outer iterations), $M$ (steps per iteration), $\mu$ (GRPO updates), $\epsilon$ (clipping), $\beta$ (KL coefficient)

**Ensure:** Fine-tuned policy $\pi_\omega$

1:  $\pi_\omega \leftarrow \pi_\omega^0$
2: **for** $n = 1, \dots, N$ **do**
3:    $\pi_{\text{ref}} \leftarrow \pi_\omega$
4:    **for** $m = 1, \dots, M$ **do**
5:       Sample minibatch $\mathcal{B} \subset \mathcal{D}$
6:       $\pi_\omega^{old} \leftarrow \pi_\omega$                                     ▷ Update old policy
7:       **for** each $q \in \mathcal{B}$ **do**
8:          Generate $G$ outputs $\{o_i\}_{i=1}^G \sim \pi_\omega^{old}(\cdot|q)$
9:          **for** $i = 1, \dots, G$ **do**
10:            Compute VERA reward:

$$R(o_i) = \lambda_1 R_{format}(o_i) + \lambda_2 R_{acc}(o_i) + \lambda_3 R_{rsn}(o_i) + \lambda_4 R_{logic}(o_i)$$

11:          **end for**
12:          Normalize rewards: $\tilde{R}(o_i) = (R(o_i) - \text{mean}(R))/\text{std}(R)$
13:          Set advantages: $\hat{A}_{i,t} \leftarrow \tilde{R}(o_i), \; \forall t \in o_i$
14:       **end for**
15:       **for** $u = 1, \dots, \mu$ **do**
16:          Update $\pi_\omega$ with gradient coefficient:

$$GC(q, o, t) = \hat{A}_{i,t} + \beta \left( \frac{\pi_{\text{ref}}(o_{i,t}|o_{i,<t})}{\pi_\omega(o_{i,t}|o_{i,<t})} - 1 \right)$$

17:       **end for**
18:    **end for**
19: **end for**
20: **return** $\pi_\omega$

---

| Model | #Params | w/ CoT | w/o CoT ($\Delta$) |
|---|---|---|---|
| Mono-InternVL | 2B | 10.7 | 22.5 (+11.8) |
| Ovis2 | 2B | 26.8 | 36.3 (+9.5) |
| InternVL2.5 | 2B | 33.3 | 33.6 (+0.3) |
| Qwen2.5-VL | 3B | 36.2 | 41.8 (+5.6) |
| Phi-3.5 | 4B | 22.1 | 33.1 (+11.0) |
| Qwen2-VL | 7B | 50.0 | 49.6 (-0.4) |
| LLaVA1.6 | 7B | 29.0 | 30.2 (+1.2) |
| InternLM-XComposer-2.5 | 7B | 32.6 | 32.6 (+0.0) |
| Qwen2.5-VL | 7B | 45.8 | 48.1 (+2.3) |
| Idefics2* | 8B | 33.3 | 36.3 (+3.0) |
| MiniCPM-V2.5* | 8B | 35.8 | 40.4 (+4.6) |
| MiniCPM-V2.6* | 8B | 38.9 | 45.0 (+6.1) |
| InternVL2* | 8B | 47.9 | 53.1 (+5.2) |
| InternVL3 | 8B | 47.7 | 50.7 (+3.0) |
| Qwen-VL-Chat* | 9B | 34.0 | 34.3 (+0.3) |
| GLM-4V* | 9B | 49.1 | 50.3 (+1.2) |

Table 4: Accuracy of **Small** ($\leq$ 15B) MLLMs with and without CoT prompting on the CII-BENCH benchmark. $\Delta$ = (w/o CoT – w/ CoT). *Results reported from (Zhang et al., 2025).

| Model | #Params | w/ CoT | w/o CoT ($\Delta$) |
|---|---|---|---|
| CogVLM2 | 19B | 22.2 | 20.3 (-1.9) |
| CogVLM2-Chinese-Chat* | 19B | 42.6 | 43.4 (+0.8) |
| Gemma3 | 27B | 39.1 | 39.1 (+0.0) |
| Qwen2.5-VL | 32B | 53.7 | 56.2 (+2.5) |
| LLaVA1.6* | 34B | 44.5 | 46.0 (+1.5) |
| InternVL3 | 38B | 52.8 | 52.7 (-0.1) |
| InternVL2* | 40B | 57.6 | 57.9 (+0.3) |
| LLaVA1.6* | 72B | 45.3 | 48.0 (+2.7) |
| Qwen2-VL* | 72B | 62.1 | 64.4 (+2.3) |
| InternVL2* | 76B | 52.6 | 52.9 (+0.3) |

Table 5: Accuracy of **Large** ($> 15$B) MLLMs with and without CoT prompting on the CII-BENCH benchmark. $\Delta$ = (w/o CoT – w/ CoT). *Results reported from (Zhang et al., 2025).

Table 6: Prompt used in Step 1 (Subgoal Planning) of the MoCoT pipeline.

You will be shown an image and a related question. Do not attempt to answer the question. Instead, analyze the question in the context of the image, focusing on what makes it semantically complex, ambiguous, or rich in interpretation. Then, decompose it into a small number of **independent sub-questions** (up to 4), each targeting a different aspect that would help a model better understand and reason about the original question. These sub-questions should:

- Reflect the implicit knowledge, reasoning steps, or background assumptions required to fully understand the question and its connection to the image.

- Avoid directly referencing answer options.

- Be useful for guiding deeper interpretation of the comic's meaning, narrative structure, symbolism, or emotional content.

Format your output as:

```
{
  "cot": "<your step-by-step reasoning about the question and how
      the image informs it>",
  "sub_questions": [
    "<sub-question 1>",
    "<sub-question 2>",
    "... (up to 4)"
  ]
}
```

Only output the JSON. You may refer to the image to inform your reasoning, but do not describe the image or answer the question.

Table 7: Prompt used in Step 2 (Localized Execution) of the MoCoT pipeline.

You will be shown an image and a related question. Your task is to answer the question using a Chain of Thought (CoT) approach, grounded in the image.

- First, examine the image and identify relevant visual elements (e.g., objects, characters, expressions, actions, spatial relationships).
- Then, reason through the question step by step, referencing the image as needed.
- Finally, answer the question directly.

Your output must be in the following JSON format:

```
{
  "cot": "<your step-by-step reasoning based on the image and
      question>",
  "answer": "<final answer here>"
}
```

Now, analyze the image and answer the question with a clear CoT reasoning process in the specified JSON format.

| Standard CoT |
| --- |

The image depicts a comparison between two individuals working on computers in 1980 and 2013. The cartoon humorously illustrates the rapid advancement of technology over the decades. While the 1980s individual struggles with the outdated equipment, the 2013 individual effortlessly handles the advanced technology. This contrast highlights the generational gap in adapting to new technologies and the challenges faced by older individuals in keeping up with modern advancements. The cartoon also suggests that the rapid evolution of technology has led to a significant increase in electronic waste, which can be harmful to the environment.

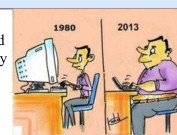

| MoCoT |
| --- |

**Sub-Question Decomposition**
What are the visible contrasts in the physical appearance and posture of the individual from 1980 to 2013, and how do they relate to lifestyle changes due to technology?
How does the progression of technology between the two years reflect societal shifts in both product design and user engagement?
What implications does the image suggest regarding the balance between technological advancement and personal health?
In what ways might this image represent generational differences in familiarity and comfort with technology?

**Sub-Question Reasoning**
The visible contrasts in the physical appearance and posture of the individual from 1980 to 2013, and how they relate to lifestyle changes due to technology, include the individual's increased weight and slouched posture in 2013 compared to the slimmer, upright posture in 1980. This reflects a shift towards more sedentary lifestyles and increased screen time associated with modern technology.
The progression of technology from 1980 to 2013 reflects a shift towards more compact and efficient designs, while societal changes have led to increased sedentary behavior and weight gain.
The image suggests that technological advancement has led to a more sedentary lifestyle, potentially contributing to health issues like obesity.
The image represents generational differences in familiarity and comfort with technology by showing the evolution of computer design and usage over time, with the character from 2013 appearing more comfortable and familiar with the technology.

**Diagnostic Thought Rationale**
The sub-questions provide insightful analysis regarding the visible contrasts in the characters from 1980 to 2013. The first answer accurately highlights the increase in weight and changes in posture due to a more sedentary lifestyle linked to technology. The second answer underscores how product design has evolved towards compactness, illustrating a societal trend toward increased screen time. The third answer supports the notion that such technological advancements have negative implications for physical health, suggesting a correlation between increased technology use and obesity. The fourth answer points out generational differences, indicating that familiarity with technology has grown but does not delve deeply into the broader implications of this comfort. Overall, the sub-questions suggest a consistent theme around the health impacts of technology and lifestyle changes, primarily pointing towards issues of obesity and sedentary behavior. However, they don't fully explore the potential humor in generational gaps, which might be significant considering one of the answer choices emphasizes this aspect.

**Final Inference Rationale**
The overall theme represented in the image reflects the consequences of technological advancement on health and lifestyle. The character from 1980 exhibits a healthier, more active posture, while the 2013 version highlights weight gain and sedentary habits fostered by modern technology...

Figure 5: Case study comparing standard CoT (Qwen-2.5-3B-VL) and our MoCoT on a visual satire task.

# F  COMPLETE QUALITATIVE COMPARISONS

In Figure 2, we highlighted three representative failure modes of Qwen-2.5VL-3B: (A) satirical target confusion, (B) symbolic misalignment, and (C) salient cue omission. In this section, we provide the complete set of responses corresponding to Figure 1, including both Qwen and ours. Moreover, we further illustrate each failure type with additional examples: satirical target confusion with Figures 15 and 16, symbolic misalignment with Figures 13 and 18, and salient cue omission with Figures 14 and 17.

Table 8: Prompt used in Step 3 (Meta-Reasoning) of the MoCoT pipeline.

---

**Two-Stage Visual Reasoning: Interpret the Deep Meaning of a Cartoon**
You are provided with:

- A **cartoon image** ('image_path')

- A **multiple-choice question** asking which of the provided options (e.g., A, B, C, . . . ) best expresses the cartoon's deep meaning

- A set of **sub-questions and sub-answers** exploring visual, symbolic, or thematic aspects of the image

Your task involves two distinct reasoning stages:
**Stage 1 — cot1: Critically Evaluate Sub-Answers** Do not try to answer the main question yet. For each sub-question and its answer:

- Assess whether the answer is accurate, coherent, visually grounded, and symbolically insightful.

- Point out strong insights (e.g., symbolism, emotional interpretation).

- Point out weak points (e.g., vagueness, factual errors, irrelevance).

Summarize in a concise paragraph or bullet list per sub-answer. The goal is to diagnose the quality of intermediate reasoning, not to solve the problem.
**Stage 2 — cot2: Independent Deep Reasoning and Final Choice (Informed by cot1)** Now interpret the cartoon from the image itself, making an independent judgment. Steps:

1. Describe the image explicitly (main objects, actions, tone, key symbols).

2. Interpret the symbolism and theme (message, human values, societal critique).

3. Compare all answer choices: explain matches and mismatches.

Finally, give your best answer.
**Final Output Format:**

```
{
  "cot1": "Your structured evaluation of the sub-answers.",
  "cot2": "Your independent reasoning and answer justification.",
  "answer": "Your final choice (e.g., A, B, C, D, or other label)"
}
```

**Example Output:**

```
{
  "cot1": "1. The answer to sub-question 1 accurately identifies the
      image's central element - a businessman climbing over others. It is
      visually grounded and symbolically points to social hierarchy.
      2. The answer to sub-question 2 misses the emotional tone -- the
          despair of those stepped on. It's a surface-level description
          without symbolic insight.
      3. Sub-answer 3 insightfully connects the broken ladder to systemic
          inequality -- a strong symbolic interpretation.",
  "cot2": "The image depicts a businessman climbing a ladder made of
      people. Those below appear crushed, while he ascends smugly. The
      exaggerated expressions emphasize exploitation. Symbolically, the
      cartoon critiques how success in capitalism often rests on the
      suffering of others.
      A: Suggests hard work pays off -- doesn't fit the exploitative
          theme.
      B: Argues society rewards the clever -- also fails to address the
          cruelty shown.
      C: Says 'one's success is built on others' pain' -- this directly
          reflects the image's symbolism.
      D: Suggests individualism is key -- irrelevant to the collective
          suffering shown. C is the best fit.",
  "answer": "C"
}
```

Table 9: Prompt used in Step 3 (Verification) of the MoCoT pipeline.

You are a logical critique model tasked with post-hoc evaluation and revision of a reasoning paragraph ('cot2') that aims to justify the selection of one of several options (e.g., A, B, C, D) in response to a visual question. **You will NOT see the image**, only the textual reasoning. **Objectives:**

1. Determine if the original 'cot2' logically supports the given final answer.

2. If it does not, return a corrected version of 'cot2'.

**Output Format:** Respond with a valid JSON object, enclosed in a markdown code block, like this:

```
{
  "Matched Answer": "A",
  "Is Consistent": true,
  "Justification": "The reasoning supports the final answer.",
  "Corrected CoT2": "The revised reasoning here."
}
```

Do not include anything outside the code block.

Table 10: Prompt used for GRPO reinforcement learning fine-tuning.

A conversation between User and Assistant. The user asks a multiple-choice question, and the Assistant solves it. The assistant first thinks about the reasoning process in the mind and then provides the user with the answer. The reasoning process and answer are enclosed within <REASONING></REASONING> and <ANSWER></ANSWER> tags, respectively, i.e., <REASONING> reasoning process here </REASONING><ANSWER> answer option label here </ANSWER>

Table 11: Prompt used for MLLM evaluation without CoT.

You are a helpful assistant for image-based reasoning. You will be given an image and a related multiple-choice question. Your task is to examine the image, understand the question and options, and then identify the correct answer.

Respond in **strict JSON format**, with only one field: - "answer": a list that contains only your final answer as a string — specifically, the correct option label (e.g., "A", "B", "C", or "D"). It should NOT include explanation or restate the option text.

**Output exactly one JSON object, and nothing else — no comments, no code blocks, no additional text.**

Here is the required format:

```
{
  "answer": ["<option label here>"]
}
```

If the answer is not certain, make your best inference and still produce a valid JSON object with your final answer.

Table 12: Prompt used for MLLM evaluation with CoT.

You will be given an image and a related multiple-choice question. Your task is to examine the image, understand the question and options, and then reason step by step before arriving at the final answer.

Respond in strict JSON format, with two fields: - "cot": a detailed step-by-step explanation showing your reasoning based on visual elements and the question and options. - "answer": a list that contains only your final answer (e.g., ["A"]).

**Output exactly one JSON object, and nothing else.**

Your output must be in the following JSON format:

```
{
  "cot": "<your step-by-step reasoning based on the image and
      question>",
  "answer": "<final answer here>"
}
```

## G  CASE STUDY ON MOCOT

To evaluate the reasoning capability of MoCoT compared to standard CoT, we analyze a cartoon that contrasts an individual using a computer in 1980 with one in 2013, as illustrated in Figure 5. The image implicitly critiques how technological progress correlates with lifestyle changes and declining physical health.

Standard CoT focuses primarily on superficial signals such as device modernization, increased user comfort, and environmental impact (e.g., e-waste). However, it fails to recognize the central theme conveyed visually—the ironic reversal between technological refinement and human physical degradation. As a result, it selects interpretations that are plausible yet miss the intended societal critique.

In contrast, MoCoT decomposes the image into modular sub-questions covering posture, body shape, product evolution, and behavioral implications. This structured process surfaces the insight that while devices become slimmer and more efficient, users become more sedentary and unhealthy. By aligning local observations with a global interpretation, MoCoT identifies the core message: modern convenience may come at the cost of physical well-being. This example demonstrates MoCoT's superiority in extracting abstract, compositional meaning from visual satire.

Table 13: Qualitative comparison on cartoon question 1.

| Image | Question |
|---|---|
| 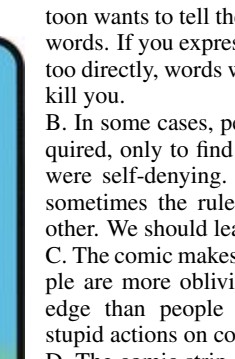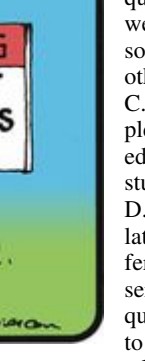 | A. As illustrated at the top of the frame, this cartoon wants to tell the viewer: be careful with your words. If you express what you think in your head too directly, words will become a sharp weapon to kill you. 
 B. In some cases, people read the warnings as required, only to find that the warnings themselves were self-denying. This can be confusing. So sometimes the rules themselves contradict each other. We should learn to judge for ourselves. 
 C. The comic makes fun of the fact that many people are more oblivious towards common knowledge than people expect, and would perform stupid actions on common and easy tasks. 
 D. The comic strip conveys a deeper meaning related to the divergent outcomes resulting from different choices and behaviors. The caricature also serves as a reminder that our actions have consequences and that making positive choices can lead to personal growth, fulfillment, and a vibrant life, while negative habits can lead to stagnation and a decline in overall well-being. |
| **Qwen2.5VL-3B** | **Ours** |
| The image shows a cartoon character standing next to a sign that reads "WARNING: DO NOT READ THIS SIGN." This sign is placed in a grassy area with a blue sky in the background. The character appears to be looking at the sign, suggesting that the viewer is meant to pay attention to the message. The sign itself is a play on words, as it is a warning sign, but the text on the sign is actually a warning against reading the sign. This creates a humorous and ironic situation where the warning is not what it seems. The deep meaning of the image can be interpreted as a commentary on the importance of being cautious with our words and actions, as the sign itself is a warning against reading it. The character's reaction to the sign, combined with the irony of the message, suggests that the viewer should be careful with theown words and actions, as they can have unintended consequences. 
 Final answer: A ✗ | The image shows a cartoon character standing next to a sign that reads "WARNING DO NOT READ THIS SIGN." This suggests a paradoxical situation where the warning itself is contradictory. The cartoon humorously highlights the tension between the warning and the instruction to not read it, implying a deeper commentary on the complexity of warnings and the human tendency to ignore or misinterpret them. The cartoon conveys a critique of how people often fail to heed clear warnings, reflecting on the self-denying nature of warnings. Option A focuses on the danger of direct communication, but it doesn't capture the cartoon's critique of warnings. Option B directly addresses the contradiction in warnings, aligning well with the cartoon's message. Option C emphasizes obliviousness to common knowledge, but it misses the core theme of the cartoon's critique. Option D discusses the consequences of choices, but it doesn't specifically relate to the cartoon's focus on warnings. Therefore, option B is the most suitable choice as it encapsulates the cartoon's deeper meaning about the self-denying nature of warnings. 
 Final answer: B ✓ |

Table 14: Qualitative comparison on cartoon question 2.

| Image | Question |
|---|---|
|  | A. Through the echo of the behavior of the mother in 1995 and the behavior of the mother in 2015, the author of the cartoon expressed that with the changes of the times, parents' educational concepts are likely to fall into a strange cycle.
B. The deep meaning conveyed by the picture relates to the unfair treatment of men and women in the workplace. It symbolizes the different challenges and obstacles faced by individuals based on their gender. The direct path represents the easier and smoother career progression often enjoyed by men, while the winding path signifies the complex and difficult journey that women often have to navigate to achieve the same level of success.
C. Our whole life is like being on a conveyor belt. We are always pursuing various goals, yet we never stop chasing them. As we grow older, we start to escape from the threat of death. This cartoon sketches out the essence of life and the dilemma of human beings through a simple scene. We are always running towards fame, fortune, power, etc., but when we reach the finish line, we may only be left with emptiness and endless anxiety.
D. The picture shows that by choosing the right way, you can reach the endpoint, while if you choose the wrong way, you will only get farther and farther away from the endpoint. |

| Qwen2.5VL-3B | Ours |
|---|---|
| The image shows two people walking on a winding path that leads to a tower. The path is marked with concentric circles, suggesting a journey or a path to a destination. Option A: This option suggests that the image is about the changes in parents' educational concepts over time. However, the image does not provide any information about parents' educational concepts or their evolution over time. It focuses more on the path and the tower, which are not directly related to this concept. Option B: This option suggests that the image is about the unfair treatment of men and women in the workplace. The winding path could symbolize the challenges faced by women, while the direct path could represent the easier career progression for men. However, the image does not provide any specific context about gender equality or workplace issues. Option C: This option suggests that the image is about the essence of life and the dilemma of human beings. The winding path could represent the pursuit of various goals, while the tower could symbolize the end or destination. However, the image does not provide any specific context about life or human dilemmas. Option D: This option suggests that the image is about the importance of choosing the right path. The winding path could represent a wrong path, while the direct path could represent the right path. This option aligns well with the visual elements of the image, where the winding path leads to a tower, suggesting a journey or a path to a destination. Based on this analysis, the most suitable deep meaning for the image is Option D: The picture shows that by choosing the right way, you can reach the endpoint, while if you choose the wrong way, you will only get farther and farther away from the endpoint.
Final answer: D ✗ | The image depicts two individuals walking on different paths, one straight and one winding, towards a destination. This suggests a metaphor for different career paths and challenges faced by men and women. The winding path represents the complexities and obstacles women often encounter, while the straight path symbolizes easier career progression for men. The cartoon conveys the theme of gender inequality in professional environments, highlighting the different challenges faced by individuals based on their gender. Option A focuses on the changes in parental educational concepts over time, which does not align with the image's focus on gender inequality. Option B directly addresses the unfair treatment of men and women in the workplace, aligning well with the image's message. Option C emphasizes the essence of life and human pursuit, but it doesn't capture the specific gender dynamics depicted in the image. Option D suggests a choice between right and wrong paths, but it misses the underlying theme of gender inequality. Therefore, option B is the most suitable choice as it encapsulates the deep meaning of the image.
Final answer: B ✓ |

Table 15: Qualitative comparison on cartoon question 3.

| Image | Question |
| --- | --- |
| ©2014 JOHN McPHERSON/DIST. BY UNIVERSAL UCLICK

EUREKA!

After years of research, Dr. Anderson devises the perfect way to get women to stop smoking: The 1,000-calorie cigarette. | A. This cartoon tells us that doing anything requires not only passion but also professional knowledge and careful preparation.
B. Calories are the amount of heat and energy contained in food, which goes against the common pursuit of women in today's times to be slim and build a good body. People tend to choose foods, drinks, etc. with low calorie content in order to lose weight, and even keep their daily calorie intake within a certain limit.
C. This picture shows that many great scientific discoveries often come from experiments with eccentricity.We should not be afraid to question and ridicule to make great discoveries.
D. The comic strip prompts reflection of the status of teachers today, and the shifting landscape of education, parental involvement, and the recognition that a student's self-confidence and personal growth should be valued alongside academic performance. |

| Qwen2.5VL-3B | Ours |
| --- | --- |
| The image depicts a humorous scenario where a scientist, Dr. Anderson, invents a 1,000-calorie cigarette as a way to get women to stop smoking. This is a satirical take on the idea that scientific discoveries can sometimes be absurd or impractical. Let's analyze each option in relation to the image: A. This cartoon tells us that doing anything requires not only passion but also professional knowledge and careful preparation. - This option is not directly related to the image. The image is more about the absurdity of the invention rather than the preparation required. B. Calories are the amount of heat and energy contained in food, which goes against the common pursuit of women in today's times to be slim and build a good body. People tend to choose foods, drinks, etc. with low calorie content in order to lose weight, and even keep their daily calorie intake within a certain limit. - This option is also not directly related to the image. The image is about a 1,000-calorie cigarette, not about calorie intake or weight loss. C. This picture shows that many great scientific discoveries often come from experiments with eccentricity. We should not be afraid to question and ridicule to make great discoveries. - This option is the most suitable deep meaning for the image. The image humorously depicts a scientist inventing a 1,000-calorie cigarette, which is an absurd and eccentric idea. This aligns with the idea that great scientific discoveries can sometimes come from unconventional and eccentric experiments. D. The comic strip prompts reflection of the status of teachers today, and the shifting landscape of education, parental involvement, and the recognition that a student's self-confidence and personal growth should be valued alongside academic performance. - This option is not related to the image. The image is about a scientist inventing a 1,000-calorie cigarette, not about the status of teachers or education. Therefore, the most suitable deep meaning for the image is option C.
Final answer: C ✗ | The cartoon depicts Dr. Anderson inventing a 1,000-calorie cigarette as a way to get women to stop smoking. This suggests a humorous take on the societal pressure to be thin and the pursuit of weight loss. The cartoon implies a critique of societal norms and the pursuit of thinness, highlighting the absurdity of creating a product to counteract these pressures. The deep meaning of the cartoon revolves around the societal expectations and the humorous commentary on the pursuit of thinness. Option A focuses on the need for passion and professional knowledge, which doesn't directly relate to the cartoon's critique of societal norms. Option B emphasizes the common pursuit of slimness, aligning well with the cartoon's critique of societal expectations. Option C discusses the nature of scientific discoveries, but it misses the satirical tone of the cartoon. Option D addresses the status of teachers and education, which is not relevant to the cartoon's message. Therefore, option B is the most suitable choice as it captures the cartoon's critique of societal expectations related to weight and thinness.
Final answer: B ✓ |

Table 16: Qualitative comparison on cartoon question 4.

| Image | Question |
|---|---|
| 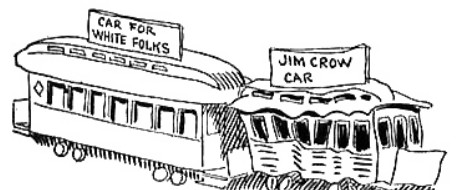 | A. The cartoon satirizes that the I.R. law is a road to nowhere, yet the government insists on enacting and enforcing it. 
 B. This cartoon satirizes many people in society who hope for change, but every one of them refuses to change themselves. 
 C. Individuals lose their unique identities and become part of a homogeneous group. This portrayal reflects the potential loss of individuality and creativity in standardized educational systems, where diversity and personal expression may be suppressed in favor of uniformity. The comic strip highlights the societal pressure to conform and raises questions about the impact of institutional structures on personal identity and freedom of expression. 
 D. It is a sign of racism, because blacks are treated differently, not only are they not allowed to be with whites, but they are kept in worse conditions. |

| Qwen2.5VL-3B | Ours |
|---|---|
| The image depicts a train with two separate compartments labeled 'Car for White Folks' and 'Jim Crow Car,' suggesting a division based on race. This satirical representation implies that despite the government's efforts to enforce racial equality, segregation persists. The cartoon likely aims to highlight the ineffectiveness of such policies and the ongoing racial disparities in society. Therefore, the most suitable deep meaning is that the cartoon satirizes that the I.R. law is a road to nowhere, yet the government insists on enacting and enforcing it. 
 Final answer: A ✗ | The image depicts a train with separate compartments labeled "CAR FOR WHITE FOLKS" and "JIM CROW CAR," suggesting segregation and discrimination based on race. This imagery conveys a critique of racial inequality and the systemic racism embedded in societal structures. The cartoon highlights the harsh realities of racial segregation, reflecting the deep-seated issues of discrimination and the loss of individuality due to societal norms. Option A focuses on the I.R. law, which doesn't directly relate to the cartoon's theme of racial discrimination. Option B emphasizes societal change, but it doesn't capture the specific critique of racial discrimination depicted in the cartoon. Option C discusses the loss of individuality in educational systems, which is not directly relevant to the cartoon's focus on racial segregation. Option D accurately encapsulates the cartoon's critique of racism and segregation, aligning perfectly with the image's message. Therefore, option D is the most suitable choice as it directly addresses the cartoon's deep meaning of racism. 
 Final answer: D ✓ |

Table 17: Qualitative comparison on cartoon question 5.

| Image | Question |
|---|---|
| 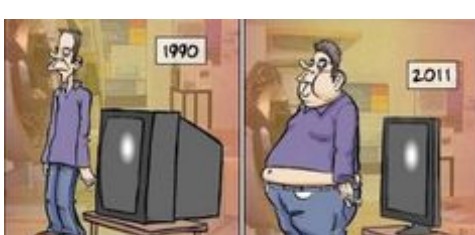 | A. The moral of this cartoon is that people's physical health is affected by their daily eating habits; on a deeper level, people's thoughts are also related to what people usually read, and what we read and come into contact with will affect us in turn.
B. The comic strip implicitly raises concerns about the impact of technological advancements on human health and lifestyle. We should change relationship between technology and human bodies, highlighting the importance of prioritizing physical health and balance in an increasingly digital age.
C. Electronic devices like TV confine humanity. We should get rid of electronic devices and embrace nature.
D. This cartoon satirizes many people in society who hope for change, but every one of them refuses to change themselves. |
| **Qwen2.5VL-3B** | **Ours** |
| The image shows a cartoon of a man in 1990 standing next to a small television, while in 2011, the same man is much larger and overweight, standing next to a much larger television. This visual progression suggests a correlation between the size of the television and the man's weight gain over time. The moral of the cartoon can be interpreted as a commentary on the impact of technology, particularly television, on human health and lifestyle. The larger television in 2011 could symbolize the increased consumption of media and the sedentary lifestyle that often accompanies it, leading to weight gain. Therefore, the most suitable deep meaning for the image is that people's physical health is affected by their daily eating habits; on a deeper level, people's thoughts are also related to what people usually read, and what we read and come into contact with will affect us in turn.
Final answer: A ✗ | The image shows a person in 1990 standing next to a small television, while in 2011, the same person is much larger and overweight, standing next to a much larger television. This visual metaphor suggests a correlation between increased screen time and physical health issues. The cartoon highlights how technological advancements have influenced human behavior and health. The deeper meaning touches on the impact of daily habits and the relationship between technology and human health. Option A focuses on the impact of eating habits, which aligns with the cartoon's theme but doesn't fully capture the broader technological impact depicted. Option B emphasizes the impact of technology on health and lifestyle, which resonates well with the cartoon's message.Option C suggests a rejection of electronic devices, which misses the cartoon's critique of technology.Option D addresses societal change but doesn't directly relate to the cartoon's core message about technology and health. Therefore, option B is the most suitable choice as it encapsulates the cartoon's deeper meaning about technology's influence on health.
Final answer: B ✓ |

Table 18: Qualitative comparison on cartoon question 6.

| Image | Question |
|---|---|
| 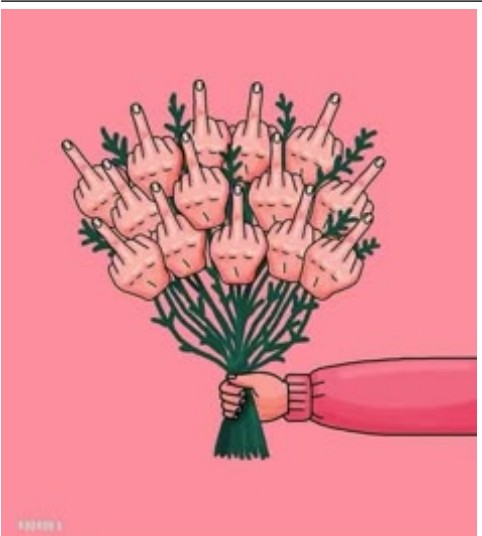 | A. This picture satirizes some people who talk too much when they give a speech, but what they actually say is illogical and ridiculous. B. If the person who sends flowers has ill intentions, then the flowers they send will only be offensive to people. No matter how well they are packaged, ill will is always hurtful. C. With flowers in your heart, your mood will naturally get better. We should be able to adjust our mood. D. This picture uses a sarcastic attitude to express the problem of people nowadays who only talk but don't take action. Without people who actually do things, all the talk from the critics and talkers is meaningless. |

| Qwen2.5VL-3B | Ours |
|---|---|

The image shows a hand holding a bouquet of flowers with middle fingers pointing upwards. This visual metaphor is often used to convey a message of defiance or rebellion. Let's analyze each option in relation to this image: A. This picture satirizes some people who talk too much when they give a speech, but what they actually say is illogical and ridiculous. This option suggests that the image is a critique of people who speak excessively but have no substance. However, the image does not seem to focus on the content of the speech but rather on the gesture of the middle fingers. B. If the person who sends flowers has ill intentions, then the flowers they send will only be offensive to people. No matter how well they are packaged, ill will is always hurtful. This option implies that the flowers themselves are offensive due to the gesture of the middle fingers. However, the flowers are not the focus of the image; the gesture is. C. With flowers in your heart, your mood will naturally get better. We should be able to adjust our mood. This option suggests a positive message about adjusting one's mood with flowers. However, the image does not convey this message; it uses the gesture of the middle fingers to make a point. D. This picture uses a sarcastic attitude to express the problem of people nowadays who only talk but don't take action. Without people who actually do things, all the talk from the critics and talkers is meaningless. This option aligns with the image's message. The gesture of the middle fingers, often associated with rebellion or defiance, can be interpreted as a critique of people who talk a lot but do little. The flowers, while not the main focus, can be seen as a symbol of the speaker's intentions, which are not well-received.
Final answer: D ✗

The image depicts a bouquet of flowers with middle fingers pointing upwards, symbolizing a gesture of defiance or rebellion. This visual conveys a sense of rebellion against conventional norms, suggesting a critique of superficial gestures. The flowers, typically associated with love and affection, are juxtaposed with the aggressive gesture, highlighting the contrast between outward expressions and underlying intentions. The deep meaning of the image likely reflects the critique of superficial acts of communication and the underlying ill intentions behind such gestures. Option A focuses on the critique of speech, but it doesn't directly relate to the aggressive gesture in the image. Option B emphasizes the offensive nature of the gesture, aligning well with the image's message. Option C suggests a positive outlook on flowers, which doesn't resonate with the aggressive symbolism. Option D addresses the problem of talk without action, but it misses the specific critique of superficial gestures depicted in the image. Therefore, option B is the most suitable choice as it encapsulates the image's critique of ill intentions behind gestures.
Final answer: B ✓

