# OpenReview forum: "Towards Faithful Reasoning in Comics for Small MLLMs"
_ICLR.cc/2026/Conference — ICLR 2026 Conference Withdrawn Submission_

### Official Review · Reviewer_tjsb · 2025-10-18

**Soundness:** 3
**Presentation:** 3
**Contribution:** 2
**Rating:** 4
**Confidence:** 4

**Summary:**

This paper investigates why Chain-of-Thought (CoT) prompting surprisingly degrades performance on comic-based visual question answering (CVQA) tasks, particularly for small multimodal large language models (MLLMs). The authors provide theoretical and empirical analysis attributing this failure to state entanglement, spurious transitions, and exploration inefficiency. They propose MoCoT (Modular Chain-of-Thought), a framework that decomposes comic reasoning into typed sub-questions (VISUAL, SYMBOLIC, NARRATIVE) followed by meta-reasoning and verification. Combined with GRPO-based reinforcement learning using a novel VERA reward, their 3B model achieves 10.4% average improvement over state-of-the-art on three comic VQA benchmarks and outperforms larger 7B models.

**Strengths:**

Novel and Important Observation: The finding that standard CoT degrades performance in CVQA is counterintuitive and practically important, especially given the widespread deployment of small MLLMs in resource-constrained settings.
Comprehensive Framework Design: MoCoT's plan-execute-verify pipeline with typed reasoning modules is well-motivated by the compositional nature of comic understanding. The integration with GRPO and structured rewards (VERA) demonstrates thoughtful system design.


Strong Empirical Results:

Consistent improvements across three benchmarks (DeepEval, YesBut, CII-Bench)

The 3B model outperforming 7B baselines is particularly impressive

Backbone-agnostic evaluation (Table 2) shows +12.1% average improvement, demonstrating generalizability

Thorough ablation studies validate each component's contribution


Qualitative Analysis: The identification of three specific failure modes (satirical target confusion, symbolic misalignment, salient cue omission) with concrete examples provides valuable insights into comic understanding challenges.

**Weaknesses:**

1. Theoretical Rigor and Validation

The theoretical framework (Section 2.1.1, Appendix B) makes several strong assumptions without empirical validation:

State factorization: The decomposition z_u = (z_u^perc, z_u^abs) assumes a clean separation between perceptual and abstract variables, but comics often blend these (e.g., visual metaphors). How natural is this factorization for real comic understanding?

Weak coupling assumption (Definition 2.2): The claim that D_KL(p(τ^(i) | τ^(j), I) || p(τ^(i) | I)) ≤ ε requires validation. What is ε empirically? Does this hold for your MoCoT decompositions?

Gap between theory and practice: The formal model assumes discrete state spaces Z with specific transition dynamics, but the actual implementation uses LLM-generated natural language rationales. How do you reconcile these?

Specific question: Can you provide empirical measurements of the coupling coefficient ε from your generated MoCoT data?

2. Reliance on Proprietary Models

A critical reproducibility concern:

MoCoT generation uses GPT-4o-mini for planning, execution, and meta-reasoning (Tables 6-9)

Only diverse sub-answer generation uses Qwen2.5-VL-7B

This creates dependency on closed-source APIs for data generation

Questions:

What happens if all MoCoT components use open-source models (e.g., Qwen2.5-VL-7B)?

What is the total cost (API calls, time) to generate 930 MoCoT trajectories?

Could you provide ablations comparing GPT-4o-mini vs. open-source model performance for each MoCoT component?

3. Limited Baseline Comparisons

The paper only compares against naive CoT and direct answering:

Missing comparisons: Self-consistency (Wang et al.), Tree-of-Thought, other structured reasoning approaches

No analysis of cost-performance tradeoffs: MoCoT requires multiple LLM calls; how does it compare to simpler approaches when controlling for computational budget?

4. VERA Reward Design Concerns

The structured reward formulation lacks sufficient justification:

Hyperparameter selection: Weights λ₁=0.05, λ₂=0.6, λ₃=0.2, λ₄=0.15 appear manually tuned without sensitivity analysis

Conditional activation: R_rsn is only activated when R_acc=1, which could create optimization difficulties (sparse signal)

Implementation details: The logic consistency reward R_logic relies on symbolic checker V, but specifics are unclear (Table 9 shows it checks "entailment" but the exact mechanism isn't described)

Request: Please provide ablations showing performance with different reward weights and analyze sensitivity.

5. Evaluation Limitations

Multiple-choice only: All benchmarks use MCQ format, which may not fully assess reasoning quality. Have you considered open-ended generation tasks?

No human evaluation: The paper claims to produce "faithful" reasoning but doesn't validate this with human judgments

Limited failure analysis: While qualitative examples show improvements, there's no systematic analysis of remaining failure modes or error categories

6. Data Construction and Generalization

Small training set: Only 745 samples for GRPO fine-tuning seems very limited. Did you explore scaling?

Domain specificity: All experiments focus on comics/cartoons. Does the approach generalize to other multimodal reasoning domains (e.g., scientific diagrams, infographics)?

Potential data leakage: You state "CII-BENCH dataset is used exclusively for validation" but earlier mention using 80% of DeepEval and YesBut training sets. Please clarify the exact data splits.

**Questions:**

Can you validate that your MoCoT decompositions satisfy the weak coupling assumption? What is the empirical KL divergence between sub-trajectories?

How sensitive is performance to the VERA reward weights? Please provide a sensitivity analysis.

What is the total computational cost (tokens, time, money) for:

Generating 930 MoCoT training examples

GRPO fine-tuning

Inference time per question compared to baseline



Have you evaluated on any open-ended comic understanding tasks with human evaluation of reasoning faithfulness?

Can you compare against self-consistency, Tree-of-Thought, or the LAD framework experimentally?

What happens if you replace GPT-4o-mini with open-source models throughout MoCoT generation?

---

### Official Review · Reviewer_FXzr · 2025-10-25

**Soundness:** 3
**Presentation:** 3
**Contribution:** 2
**Rating:** 4
**Confidence:** 4

**Summary:**

This paper tries to address the problem of why CoT prompting degrades the performance of small MLLMs on Comic Visual Question Answering. The authors theoretically attribute this failure to state entanglement, spurious transitions, and exploration inefficiency. To solve this, the paper proposes a two-stage framework: 1) generating high-quality, decomposed reasoning data using a Modular Chain-of-Thoughtpipeline, and 2) fine-tuning the model using a novel structured reward and the GRPO reinforcement learning algorithm. Experiments show the method significantly outperforms baselines on three CVQA benchmarks and demonstrates generalizability as a plug-in component for various small MLLMs.

**Strengths:**

1. The paper correctly identifies and analyzes the failure of standard CoT in the specific and challenging domain of CVQA, which is particularly relevant for the deployment of small models.
2. The MoCoT design aligns with how humans might parse comics, and the VERA reward function is comprehensive in its consideration of multiple facets of reasoning.
3. The method achieves SOTA performance with a 3B model (even surpassing 7B baselines) and shows an average 12.1% improvement as a plug-in across multiple models, proving its effectiveness and generality.

**Weaknesses:**

1. The VERA reward function consists of four weighted components ($\lambda_1$ to $\lambda_4$), and these weights are empirically chosen. The lack of a sensitivity analysis or ablation study on these hyperparameters makes it unclear how optimal this design is or if the model is highly sensitive to these specific values.
2. While the title emphasizes small MLLMs, the experiments are heavily focused on a 3B model, supplemented by 2B and 4B models. It is unclear how this method generalizes to even smaller models (e.g., <1B) or slightly larger ones (e.g., 13B). The definition and discussion of "small" could be more thorough.
3. The paper is evaluated on three specialized CVQA benchmarks. While these are relevant datasets, they are all relatively new or specific. It remains unknown if the framework's benefits would transfer to other multimodal tasks requiring symbolic reasoning, such as scientific charts or general VQA.
4. The paper lacks a deep analysis of where MoCoT itself might fail. In what cases does the "plan-execute-verify" process also break down? For instance, can the planner generate flawed sub-questions?
5. The paper uses GRPO as the RL algorithm, justifying it by its lack of need for an explicit value function. However, a comparison to more standard and widely-used RLHF algorithms is missing, making it hard to judge if GRPO is a key factor in the success.
6. The title stresses "Faithful Reasoning," which the VERA reward pushes for via $R_{rsn}$ and $R_{logic}$. However, the evaluation relies primarily on final answer accuracy. The claim of faithfulness would be stronger with more direct evaluations, such as human assessment of the generated reasoning chains' plausibility.

**Questions:**

1. Typos: There is a '1' at Line 309-310
2. See weakness

---

### Official Review · Reviewer_Dksn · 2025-10-28

**Soundness:** 2
**Presentation:** 2
**Contribution:** 1
**Rating:** 2
**Confidence:** 4

**Summary:**

The paper studies why standard Chain-of-Thought (CoT) prompting underperforms for comic-based visual QA (CVQA)—especially in small MLLMs—and introduces a two-part solution: MoCoT, a modular “plan–execute–verify” reasoning pipeline that decomposes problems into VISUAL / SYMBOLIC / NARRATIVE subgoals, and VERA, a structured reward used with GRPO reinforcement fine-tuning to encourage verifiable, coherent reasoning.

**Strengths:**

1. Sound problem decomposition: a typed subgoal planner → local executors → meta-reasoner + symbolic checker, aligning with the domain’s perceptual vs. abstract reasoning split.
2. Demonstrates that structure beats scale in this domain: a 3B model with the proposed module rivals or surpasses larger baselines on DEEPEVAL and remains competitive on YESBUT/CII-Bench.

**Weaknesses:**

1. The proposed model is trained and fine-tuned on comic data (including training splits), while baselines are evaluated zero-shot; this inflates reported gains.
2. All benchmarks are comic-based; no tests on general visual-CoT domains like math, chart, or game reasoning, despite claims of broader applicability.
3. Theoretical claims rely on unverifiable assumptions (e.g., disjoint type spaces, weak coupling, perfect verifier), making the proofs weakly grounded.
4. The paper omits comparisons with modern visual-CoT or RL-CoT systems that use multi-agent or self-verification methods.

**Questions:**

1. VERA weights are set empirically (0.05,0.6,0.2,0.15). How robust are gains to these choices?
2. What is the accuracy of subgoal typing (VISUAL / SYMBOLIC / NARRATIVE)? When typing is wrong, does the verifier catch the resulting inconsistencies, or do errors silently propagate?
3. Out-of-domain portability. Add a pilot experiment on a related but distinct domain (e.g., meme VQA or satirical news cartoons), keeping training the same, to substantiate the “structure over scale” claim.
4. The paper introduces a structured GRPO reward (VERA), but no explicit ablation is shown for the 4 kinds of rewards.

---

### Official Review · Reviewer_eNW2 · 2025-10-30

**Soundness:** 2
**Presentation:** 3
**Contribution:** 2
**Rating:** 4
**Confidence:** 5

**Summary:**

This paper addresses the issue of unfaithful reasoning in comic-based visual question answers (CVQA) for small multimodal large language models (MLLMs). The authors identify that standard Chain-of-Thought (CoT) degrades performance due to state entanglement, spurious transitions, and inefficient exploration. To solve this problem, they propose a two-stage framework combining Modular Chain-of-Thought (MoCoT) and Verified Enhanced RewArd (VERA). MoCoT generates high-quality rationales using a plan-execute-verify process that decomposes questions into visual, symbolic, and narrative modules. Then, the small MLLMs are fine-tuned using Group Relative Policy Optimization (GRPO) guided by VERA from format, accuracy, similarity, and logic dimensions. Experiments across three CVQA benchmarks show that the method improves accuracy and enables the 3B model to outperform the 7B models, establishing more faithful, interpretable reasoning.

**Strengths:**

- The paper systematically formalizes state entanglement, spurious transitions, and exploration inefficiency problems to explain why standard CoT harms small MLLMs more severely in the CVQA domain.
- The plan–execute–verify pipeline enables a task-aligned reasoning process for CVQA, allowing language models to decompose complex comic questions. This structured design has the broader potential to enhance their ability to capture challenging human humor, satire.
- The paper is well structured and easy to follow. The figures and cases help provide intuition.

**Weaknesses:**

- The VERA reward weights are empirically chosen. Without sensitivity analysis, it is unclear how robust the improvements are to these choices.
- The paper emphasizes faithfulness in comic reasoning, but the evaluation only focuses on accuracy. Faithfulness should be directly assessed to support the claim of faithful reasoning.
- The VERA reward is described conceptually. Each component lacks explicit formulas or operational definitions. Even if the descriptions are reasonable, the absence of precise mathematical specifications prevents assessing the validity of the design.
- While Table 3 shows module-level ablation, there is no analysis on each VERA component. Their necessity and interaction remain unexplored.
- The paper claims to identify the specific state entanglement, spurious transitions and inefficient exploration in comic reasoning. However, the first two phenomena resemble the common weak vision–text alignment and hallucination challenges in multimodal reasoning. The paper does not sufficiently justify why these problems are unique to comic reasoning, and they are different from standard multimodal reasoning.
- The paper does not specify how baselines are implemented or prompted, leaving the fairness and reproducibility of baseline comparisons unclear.

**Questions:**

- Different cultures may have different types of satires. Could the authors discuss whether cultural differences in satire and comic styles influence model reasoning performance? It would be valuable to know whether the model performs differently on culture-specific cases or particular types of satire.
- Please provide an ablation study isolating each VERA sub-reward and a sensitivity analysis of the weights to discuss the necessity and contributions of each component.
- Since faithful reasoning is the central claim of the paper, please provide a clear and formal definition of this concept and a direct quantitative metric to evaluate it.
- Could the authors clarify why the three identified issues are unique to comic reasoning rather than general challenges in multimodal reasoning?
- Given that the training set contains only 745 samples, could the authors evaluate the model’s generalization reasoning ability to confirm that the model’s improvements are not due to overfitting on the small training set?
- Could the authors provide the detailed prompt templates and implementation details for the baseline models with and without CoT?

---

### Note · Authors · 2025-12-20

I have read and agree with the venue's withdrawal policy on behalf of myself and my co-authors.